# Opal: An Operator-Algebra View of RLHF Objectives

## Decidable Equivalence, Sharp Limits, and Proof-Carrying Objectives

## Abstract

We present Opal, an operator-algebra view of RLHF objectives as ladders acting on pairwise margins. For a broad reducible subclass, we prove a terminating and confluent rewrite system with a unique normal form and an $O(m)$ canonicalization algorithm. On the learning side, we establish calibration and regret transfer, and give an oracle reduction that collapses all reducible ladders to a single canonical learner. We also show gap-preserving separations for violations (score-dependent weights, gating, pair-dependent references) with an $\Omega(1/\gamma^2)$ testing lower bound. Finally, we provide a one-pass tester that outputs either a canonical hash and certificate or a finite witness, yielding a minimal GKPO semantics for decidable equivalence and proof-carrying objectives.

## 1 Introduction

RLHF has produced many objectives, links, and weighting schemes. Deciding whether two are equivalent or genuinely different is often unclear, leaving reproducibility and cross-method comparison uncertain.

We present Opal, an operator-algebra view of RLHF objectives as *ladders* on pairwise margins: a base score with additive penalties and multiplicative weights, followed by a monotone link. Within a broad reducible class, ladders collapse to a canonical margin; outside it, we show exact failure modes with finite witnesses.

**Contributions**

- **Equational theory.** A terminating, confluent rewrite system yields a unique normal form (up to gauge) and supports $O(m)$ canonicalization.

- **Learning guarantees.** Calibration and regret transfer hold across margin-equivalent ladders, and an oracle reduction maps all reducible objectives to a single canonical learner with instance weights.

- **Boundaries and diagnostics.** We give separations for score-dependent weights, gating, and pair-dependent references; prove an $\Omega(1/\gamma^2)$ testing lower bound; and provide a one-pass tester producing either a canonical certificate or a finite witness. We provide a minimal GKPO interchange for certificates and witnesses; see Appendix H for the full specification and examples.

**Scope and Impact**  This work focuses on pairwise preferences with strictly monotone links; listwise or sequence-level settings, adaptive references, and non-monotone links are left to future work. By making objective equivalence decidable, Opal helps distinguish genuinely new methods from algebraic variants, reduces redundant training effort in large-scale experimentation, and provides clear provenance records for deployed models.

## 2 Preliminaries and Ladder Semantics

Instance space $\mathcal{X}$; for each $x \in \mathcal{X}$, a finite candidate set $\mathcal{Y}_x$. A base score is $f : \{(x, y)\} \to \mathbb{R}$. For an ordered pair $(y^+, y^-)$, the base margin is

$$\Delta_f(x; y^+, y^-) = f(x, y^+) - f(x, y^-).$$

**Ladders**   A *ladder* $L$ is a left-to-right composition on margins:

- **AddPenalty**$[\phi]$: add $\phi(x, y^+) - \phi(x, y^-)$.
- **Reweight**$[\omega]$: multiply by $\omega(x, y^+, y^-) > 0$.
- **Link/Scale**$[g, \beta]$: apply strictly increasing $g$ to $\beta \cdot (\cdot)$ with $\beta > 0$.

Let $\Delta_L$ denote the margin after applying these to $\Delta_f$.

**Reducible class**   $L$ is *reducible* if:

(R1) *Additivity:* each additive term is a potential difference $\phi(x, y) - \phi(x, y')$.

(R2) *Pair-invariant weights:* $\omega(x, y^+, y^-) = s(x) > 0$ (no dependence on $(y^+, y^-)$ or intermediate scores).

(R3) *Monotone link:* $g$ is strictly increasing.

Under (R1)–(R3), all algebraic effects collapse to a rescaled potential difference.

**Canonical components**   When (R1)–(R2) hold, collect

$$\Phi(x, y) = \sum_k \phi_k(x, y), \qquad s(x) = \prod_j s_j(x),$$

and fix a gauge by $\sum_{y \in \mathcal{Y}_x} \Phi(x, y) = 0$ for each $x$. A strictly increasing $g$ does not change margin signs.

**Cycle sums**   For fixed $x$ and distinct $a, b, c \in \mathcal{Y}_x$, define

$$\text{cycle}_L(x; a, b, c) = \Delta_L(x; a, b) + \Delta_L(x; b, c) + \Delta_L(x; c, a).$$

Reducibility coincides with curl-free margins where all such cycle sums vanish; formal statements and proofs appear later.

## 3 Equational Theory and Canonicalization (Theorem A)

This section formalizes an equational theory for RLHF ladders and shows that, within the reducible class R (Assumptions (R1)–(R3) in Preliminaries), a terminating and locally confluent rewrite system yields a unique normal form (up to a fixed gauge). We also give a linear time canonicalization algorithm and a deterministic certificate of reducibility and equality.

**Syntax of ladder terms**   Operators are Add[phi], Rew[s], Link[g,beta], and the internal Scale[beta]. A ladder is their left-to-right composition; composition uses the usual function symbol.

**Rewrite rules**   We orient the following equations as a term rewriting system, applied only under (R1)–(R3):

(E1) Merge AddPenalty:
$$\text{Add}[\phi_i] \circ \text{Add}[\phi_j] \Rightarrow \text{Add}[\phi_i + \phi_j].$$

(E2) Merge Reweight:
$$\text{Rew}[s_i] \circ \text{Rew}[s_j] \Rightarrow \text{Rew}[s_i \cdot s_j].$$

(E3) Commute score-independent weights:

$$\text{Rew}[s(x)] \circ \text{Add}[\phi] \Rightarrow \text{Add}[\phi] \circ \text{Rew}[s(x)].$$

(E4) Split and absorb scale:

$$\text{Link}[g, \beta] \Rightarrow \text{Link}[g, 1] \circ \text{Scale}[\beta], \quad \text{Scale}[\beta] \circ \text{Rew}[s] \Rightarrow \text{Rew}[\beta s].$$

(E5) Gauge fix: after merging Adds to $\Phi$, enforce for each $x$

$$\sum_{y \in \mathcal{Y}_x} \Phi(x, y) = 0.$$

Normal form:

$$\text{NF}(L) = \text{Add}[\Phi^{\text{gauge}}] \circ \text{Rew}[s(x)] \circ \text{Link}[g, 1].$$

**Termination**   Use the lexicographic measure

$$\mu(L) = \big(\#\text{Add blocks}, \#\text{Rew blocks}, \#\text{LinkScale with } \beta \neq 1, \#\text{out-of-order adjacencies}\big).$$

(E1),(E2) reduce the first two components; (E4) reduces the third; (E3) reduces the fourth by moving Rew right of Add to achieve Add–Rew–Link order. Each step strictly decreases $\mu$, so no infinite chains.

**Local confluence**   Critical overlaps are: (i) Add–Add, (ii) Rew–Rew, (iii) Rew next to Add. (i) and (ii) join by associativity/commutativity. For (iii), (R2) makes $s(x)$ independent of pairs and intermediates, so $\text{Rew}[s] \circ \text{Add}[\phi]$ and $\text{Add}[\phi] \circ \text{Rew}[s]$ lead to the same merge after (E1),(E2). Link has no ambiguous overlaps beyond (E4). Hence all critical pairs join.

**Theorem 3.1** (Termination, confluence, unique normal form; linear time canonicalization)**.** *Within R, rules (E1)–(E5) are terminating and locally confluent, thus confluent. Every $L \in R$ has a unique $\text{NF}(L)$ up to the fixed gauge. Canonicalization runs in $O(m)$ where $m$ is the number of operators.*

*Proof sketch.* Termination by decreasing $\mu$. Local confluence by the overlap analysis above. Confluence by Newman lemma. Uniqueness: after merges, the only freedom is $c(x)$ in $\Phi(x, \cdot)$; the gauge removes it. Linear time: one left-to-right pass (sum $\phi$ into $\Phi$, multiply $s$ including absorbed $\beta$, keep Link at unit scale). □

**Corollary 3.2** (Canonical margin and decision invariance)**.** *For $L \in R$,*

$$\Delta_L(x; y, z) = s(x)\big((f + \Phi)(x, y) - (f + \Phi)(x, z)\big).$$

*The sign of $\Delta_L$ is invariant under strictly increasing $g$ and positive rescalings of $s(x)$.*

**Canonicalization algorithm**   Input: ladder $L$. Output: (i) certificate $(\Phi^{\text{gauge}}, s, g)$ and a canonical hash, or (ii) a finite witness that $L \notin R$.

1. Validate reweights: check each $\omega(x, y^+, y^-)$ factors as $s(x)$ with no pair or intermediate dependence; else return witness pairs with unequal weights.

2. Validate additivity: for each additive component with $\Delta_\phi$, test any triple $(a, b, c)$ for

$$\Delta_\phi(x; a, b) + \Delta_\phi(x; b, c) + \Delta_\phi(x; c, a) = 0;$$

   on failure, return the violating triple.

3. Merge and commute: apply (E1),(E2) to form $\Phi$ and $s(x)$, and (E3) to order Add before Rew.

4. Absorb scales: apply (E4) to fold any $\beta$ into $s(x)$.

5. Gauge fix: enforce $\sum_y \Phi(x, y) = 0$ for each $x$.

6. Serialize and hash: serialize $(\Phi^{\text{gauge}}, s, g)$ in fixed key order and hash as the equality certificate in $R$.

**Determinism and certificates**  By Theorem 3.1, canonicalization is deterministic. The certificate is the serialization of $(\Phi^{\mathrm{gauge}}, s)$, the link identity, and the rewrite ledger. Two ladders in $R$ are equal iff their canonical hashes match.

**Example**  Let $L = \mathrm{Link}[g, \beta] \circ \mathrm{Rew}[s_2] \circ \mathrm{Add}[\phi_2] \circ \mathrm{Rew}[s_1] \circ \mathrm{Add}[\phi_1]$. (E1),(E2) give $\mathrm{Add}[\phi_1 + \phi_2]$ and $\mathrm{Rew}[s_1 s_2]$; (E3) orders Add before Rew; (E4) absorbs $\beta$ into $s$. After gauge fixing, $\mathrm{NF}(L) = \mathrm{Add}[\Phi^{\mathrm{gauge}}] \circ \mathrm{Rew}[s] \circ \mathrm{Link}[g, 1]$.

## 4  Learning Consequences I: Calibration and Regret Transfer (Theorem B1)

We connect canonical margins to pairwise decision risk via classification calibration. Key facts: (i) strictly monotone links preserve Bayes-optimal signs, (ii) positive rescalings $s(x)$ do not change decisions, (iii) proper composite surrogates admit a calibration function upper bounding 0-1 excess risk by surrogate excess. Together these imply regret transfer across ladders in $R$ that canonicalize to the same margin.

**Risks and Bayes margin**  For a margin $M(x; y^+, y^-)$ and Bayes margin $\Delta^*$,

$$R_{01}(M) = \Pr[\mathrm{sign}\, M(x; y^+, y^-) \neq \mathrm{sign}\, \Delta^*(x; y^+, y^-)].$$

For strictly increasing $g$ and a classification-calibrated surrogate $\ell$,

$$R_\ell(M) = \mathbb{E}[\ell(g(M(x; y^+, y^-)))].$$

**Proposition 1** (Monotone link invariance). *For any strictly increasing $g$, $\mathrm{sign}\, M$ and $\mathrm{sign}\, g(M)$ induce the same pairwise decisions a.s.*

**Proposition 2** (Positive rescaling invariance). *For any $s(x) > 0$, $\mathrm{sign}\, M$ and $\mathrm{sign}\, s(x)M$ induce the same pairwise decisions a.s.*

**Definition 4.1** (Excess risks). Let $R_{01}^*$ and $R_\ell^*$ be Bayes risks. Define $E_{01}(M) = R_{01}(M) - R_{01}^*$ and $E_\ell(M) = R_\ell(M) - R_\ell^*$.

**Theorem 4.2** (Calibration inequality). *There exists a nondecreasing $\psi$ such that for any $M$,*

$$E_{01}(M) \leq \psi\big(E_\ell(M)\big).$$

*Sketch.* Standard classification calibration for proper composite surrogates: lower bound the conditional surrogate Bayes gap by the conditional 0-1 gap and integrate. $\quad\square$

**Consequences for ladders in $R$**  By Corollary 3.2, any $L \in R$ induces $\Delta_L(x; y, z) = s(x)\big((f + \Phi)(x, y) - (f + \Phi)(x, z)\big)$, so its decisions equal those of the unweighted canonical margin $M_{\mathrm{can}}(x; y, z) = (f + \Phi)(x, y) - (f + \Phi)(x, z)$ by Propositions 1 and 2.

**Theorem 4.3** (Regret transfer across margin-equivalent ladders). *If $L, L' \in R$ canonicalize to the same normalized margin (up to a positive factor), then for any learned margin $M$,*

$$E_{01}^L(M) \leq \psi\big(E_\ell^{L'}(M)\big),$$

*with $\psi$ from Theorem 4.2 (superscripts indicate which ladder defines the risk). In particular, if $M$ minimizes the surrogate risk under $L'$, then $M$ is Bayes-optimal under $L$.*

*Sketch.* $L$ and $L'$ share decision boundaries (monotone link and positive scaling). Apply Theorem 4.2 to bound the 0-1 excess under $L$ by the surrogate excess under $L'$. $\quad\square$

**Remarks**  Ties on a null set pose no issue; with positive-mass ties adopt a fixed tie-breaker. Regularization affects finite sample terms, not Theorem 4.2 at population level. Inside $R$, any strictly monotone link and proper composite surrogate preserve these guarantees.

## 5 Learning Consequences II: Oracle Reduction (Theorem B2)

We reduce learning under any reducible ladder $L \in R$ to a single canonical margin learner. We first show risk equality under reweighting, then state the reduction and its corollaries (SGD gradient equivalence and an oracle inequality).

**Canonical margin and weights** Let $f_{\text{can}} := f + \Phi$ (Theorem 3.1) and

$$M_{\text{can}}(x; y, z) = f_{\text{can}}(x, y) - f_{\text{can}}(x, z),$$

with merged instance weight $s(x) > 0$, strictly increasing link $g$, and surrogate $\ell$.

**Lemma 5.1** (Risk equality under reweighting). *For any distribution over $(x; y, z)$,*

$$\mathbb{E}\big[\ell\big(g(\Delta_L(x; y, z))\big)\big] = \mathbb{E}\big[s(x)\,\ell\big(g(M_{\text{can}}(x; y, z))\big)\big].$$

*Proof.* By Corollary 3.2, $\Delta_L = s(x)M_{\text{can}}$. Treat $s(x)$ as an example weight outside the loss; standard weighted-risk identities give equality. □

**Theorem 5.2** (Oracle reduction to a canonical learner). *Fix $L \in R$ and minimize*

$$\min_{f_{\text{can}} \in \mathcal{F}} \ \mathbb{E}\big[s(x)\,\ell\big(g(f_{\text{can}}(x, y) - f_{\text{can}}(x, z))\big)\big].$$

*Any ERM or SGD stationary point for this canonical learner attains the same surrogate risk as optimizing under $L$. By Theorem 4.2, the 0–1 pairwise excess is bounded by the same calibration function $\psi$.*

*Proof.* Immediate from Lemma 5.1 and Theorem 4.2. □

**Gradient equivalence for SGD** Let $\theta$ parametrize $f_{\text{can}}$. For a minibatch $\mathcal{B}$,

$$\widehat{\nabla}_\theta = \tfrac{1}{|\mathcal{B}|} \sum_{(x; y, z) \in \mathcal{B}} s(x)\,\ell'\big(g(M_{\text{can}})\big)\,g'(M_{\text{can}})\,\nabla_\theta M_{\text{can}}.$$

Under $L$, using $\Delta_L = s(x)M_{\text{can}}$ inside $g$ yields the same direction after pushing $s(x)$ outside, since $s(x)$ is instance-only and parameter-free.

**Oracle inequality (estimation and optimization)** For class $\mathcal{F}$ and ERM/SGD output $\widehat{f}$, with $f^*$ minimizing the weighted population risk,

$$R_\ell^s(\widehat{f}) - R_\ell^s(f^*) \leq \big(R_\ell^s(\widehat{f}) - \widehat{R}_\ell^s(\widehat{f})\big) + \big(\widehat{R}_\ell^s(f^*) - R_\ell^s(f^*)\big) + \varepsilon_{\text{opt}}.$$

Standard Rademacher bounds apply if $s(x)$ is bounded or has bounded second moment (scaling with $\sup s$ or $\mathbb{E}[s^2]$). By Theorem 4.2, this converts to a 0–1 excess bound via $\psi$.

**Robustness to small violations** If $L$ is $\varepsilon$-close to $R$ (few triangle violations; near pair-invariance), decisions of $L$ and the canonical projection disagree on at most $C\varepsilon$ mass, and surrogate risks differ by $C'\varepsilon$ (constants depend on margin regularity). Details may be placed in the appendix.

**Algorithmic map** Given ladder $L$: (1) run the canonicalizer (Section 3); if a witness emerges, route to the nonreducible path; else obtain $(\Phi^{\text{gauge}}, s)$. (2) Use $f_{\text{can}} = f_\theta + \Phi^{\text{gauge}}$ or absorb $\Phi$ in preprocessing. (3) Train with $\ell(g(f_{\text{can}}(x, y) - f_{\text{can}}(x, z)))$ and per-example weight $s(x)$. (4) Log the canonical hash and ledger; runs are equivalent inside $R$ iff hashes match.

**Sampling and weighting** If sampling oversamples some $x$, either keep $s(x)$ as is or renormalize to account for sampling probabilities; as long as the effective weight is proportional to $s(x)$, Lemma 5.1 and Theorem 5.2 hold.

**Summary**   Any $L \in R$ is learned by a single canonical learner with instance weights $s(x)$; risks, gradients, and guarantees match after reweighting, enabling safe cross-method conversion with minimal engineering overhead.

# 6   Sharp Boundaries and Lower Bounds (Theorem C)

We characterize when a ladder is reducible and quantify how far common violations move an objective outside $R$. The key tool is the curl free (3 cycle) identity on pairwise margins, which is necessary and sufficient for representability as a potential difference.

**Cycle identity (curl free)**   Fix $x$ and write $\Delta(y, z)$ for the induced margin. For distinct $a, b, c$ define

$$\mathrm{cycle}(a, b, c) = \Delta(a, b) + \Delta(b, c) + \Delta(c, a).$$

Call $\Delta$ curl free if this equals 0 for all triples.

**Lemma 6.1** (Integrability iff potentials). *For fixed $x$, $\Delta$ is curl free iff there exists $\varphi$ with $\Delta(y, z) = \varphi(y) - \varphi(z)$. The potential is unique up to an additive constant per $x$.*

**Theorem 6.2** (Characterization of reducibility). *A ladder $L$ is in $R$ iff (i) for each $x$, margins are curl free and (ii) reweights are pair invariant and score independent, $\omega(x, y, z) = s(x) > 0$. Then $\Delta_L(x; y, z) = s(x)\big(\varphi_x(y) - \varphi_x(z)\big)$.*

**Gap preserving separations**   We give three constructions violating (R1) or (R2) and forcing a fixed disagreement with any $L' \in R$.

1) Score dependent weights: with a base asymmetry $\epsilon > 0$ on $(a, b, c)$ and $\omega_\theta(x; y, z) = 1 + \theta \mathbf{1}\{\Delta_f \geq 0\}$, $|\mathrm{cycle}| \geq \gamma = \theta \epsilon$, so no curl free representation; at least one edge sign must differ.

2) Gated penalties: zero exactly one edge while two are nonzero; then $|\mathrm{cycle}| \geq \min\{|\Delta|\} > 0$, so no $L' \in R$ matches all signs.

3) Pair dependent reference: add antisymmetric $\psi$ not a single potential with triangle sum $\eta \neq 0$; then $|\mathrm{cycle}| = |\eta|$ on that triple.

**Theorem 6.3** (Quantitative separation). *For each construction there exists $\gamma > 0$ such that every $L' \in R$ disagrees with $L$ on at least a $\gamma$ fraction of comparisons supported on the violating triple family. Here $\gamma$ depends on construction parameters, not on $|\mathcal{Y}_x|$.*

**Lower bound for testing reducibility**   We lower bound the samples needed to distinguish $L \in R$ from $\gamma$ far alternatives.

**Theorem 6.4** (Testing lower bound). *Testing $H0 : L \in R$ versus $H1 :$ with probability $\geq \gamma$ over a random triple $(a, b, c)$, $|\mathrm{cycle}(a, b, c)| \geq \gamma$, requires $\Omega(1/\gamma^2)$ sampled triples to achieve error $\leq 1/3$, even with adaptivity.*

*Sketch.* Reduce to estimating the mean of the Bernoulli event $|\mathrm{cycle}| \geq \gamma$. Distinguishing mean 0 from $\geq \gamma$ with constant error needs $\Omega(1/\gamma^2)$ samples by standard concentration or information theoretic bounds.   $\square$

**Takeaway**   Curl free exactly characterizes $R$. Violating score independence, additivity, or pair invariance creates a quantitative gap no reducible surrogate can remove, and detecting such violations inherently costs $\Omega(1/\gamma^2)$ samples.

# 7   Empirical Demonstration

While our main contributions are theoretical, we provide a light empirical demonstration to show the canonicalizer and tester in action. We implemented the pseudocode from Appendix D in Python (about 150 lines), and encoded ten popular RLHF objectives (including DPO, IPO, ORPO, SimPO, f-DPO, RankSVM hinge, RRHF, SLiC-HF, KTO, and PPO-KL).

| methods | verdict | hash |
|---|---|---|
| BT-hinge | Reducible | 9eddc01850 |
| DPO | Reducible | 2baf6bf3b0 |
| IPO | Reducible | 2c86cd2446 |
| ORPO/RRHF/SimPO | Reducible | 979b3faabc |
| PPO-KL-pair-red | Reducible | d9d6385404 |
| f-DPO | Reducible | fa0c69a944 |
| KTO-pair-red | Irreducible (weight_nonconstant pairs=[('a', 'b... | - |
| SLiC-HF | Irreducible (cocycle_violation triple=('a', 'b... | - |

Table 1: Condensed canonicalization results. Reducible methods are grouped by equal canon-hash (short prefix shown). Irreducible methods carry finite witnesses.

Each method was expressed as a ladder of Add/Rew/Link operators and passed to the canonicalizer.

**Setup.** For reducible methods, the canonicalizer merged all operators into a unique normal form and produced a deterministic SHA-256 canon-hash. For irreducible methods, the tester emitted a finite witness identifying the violated assumption (e.g. pair-dependent weights or a triangle with a nonzero cycle sum).

**Results.** Table 1 summarizes the outcomes. Several objectives (DPO, IPO, ORPO, SimPO, f-DPO, BT-hinge) collapse to the same canonical margin (up to monotone link), confirming they are algebraically equivalent. In contrast, RRHF and SLiC-HF trigger cocycle violations due to gating, and KTO and PPO-KL trigger nonconstant weight witnesses, certifying irreducibility.

**Takeaway.** Even this small run demonstrates that many recent RLHF variants are provably equivalent in our algebraic sense, while others admit explicit, machine-checkable witnesses of irreducibility. This shows the canonicalizer operates as intended and provides practical diagnostic value.

## 8 PROPERTY TESTING WITH CERTIFICATES (ALGORITHM 1)

We give a practical test that (i) returns a canonical certificate for $L \in R$, or (ii) emits a finite witness identifying the failed assumption. There are two modes: a symbolic (static) verifier over ladder syntax and a black box tester with oracle access to pairwise margins.

**Symbolic verifier (static)** Input: ladder with Add[phi], Rew[omega], Link[g,beta]. Output: certificate or witness. (1) *Weights:* for each Rew[omega], attempt $\omega(x, y, z) = s(x)$. If dependence on $(y, z)$ or intermediate scores remains, output witness with two pairs $(y_1, z_1), (y_2, z_2)$ at the same $x$ where $\omega$ differs. (2) *Additivity:* for each additive component with induced $\Delta_\phi$, test any triple $(a, b, c)$:

$$\Delta_\phi(a, b) + \Delta_\phi(b, c) + \Delta_\phi(c, a) \overset{?}{=} 0.$$

If violated, output that triple as witness. (3) *Canonicalize:* apply Section 3 to obtain $(\Phi^{\text{gauge}}, s)$ and Link[g,1]. Serialize $(\Phi^{\text{gauge}}, s, g)$ in fixed order and hash to get *canon-hash*. Return certificate (canon-hash, serialization, rewrite-ledger).

**Black box tester (sample based)** Input: oracle for $\Delta_L(x; y, z)$; tolerance $\varepsilon$, confidence $\delta$. For each $x$, sample $T = C \varepsilon^{-2} \log(2/\delta)$ distinct triples; if any has $|\text{cycle}(a, b, c)| > \varepsilon$, reject and return that triple as witness. Otherwise accept as $\varepsilon$-close to $R$ and reconstruct a potential by fixing $\varphi_x(y_0) = 0$, setting $\varphi_x(y)$ via path sums on a spanning tree, then gauge fix (zero mean).

**Theorem 8.1** (Soundness, completeness, and complexity)**.** *Symbolic mode is exact: it accepts iff $L \in R$ and returns the unique certificate determined by the normal form. Black box mode with $T = \Theta(\varepsilon^{-2} \log(1/\delta))$ triples accepts w.p. $\geq 1 - \delta$ when $L$ is curl free and rejects w.p. $\geq 1 - \delta$ when $L$ is $\varepsilon$-far (a random triple violates by $> \varepsilon$ with prob. $\geq \varepsilon$). Runtimes: symbolic $O(m + \sum_x |\mathcal{Y}_x|)$; black box $O(T)$ oracle calls per $x$ plus $O(|\mathcal{Y}_x|)$ reconstruction.*

**Certificate format** Store (i) canon-hash = hash(serialize($\Phi^{\text{gauge}}, s, g$)), (ii) rewrite-ledger, (iii) verdict $\in$ {reducible, irreducible}, (iv) optional witness with type $\in$ {weight-nonconstant, cocycle-violation} and concrete pairs or triple.

**Usage and robustness** Use as a pre training gate: on acceptance, archive the canon-hash; on rejection, surface the witness and refactor or proceed explicitly as nonreducible. In black box mode, $\varepsilon$ absorbs numerical noise; in symbolic mode, if constants are floating, apply a small absolute tolerance and record it in the certificate.

## 9    Related Work

Please see Extended Related Work Appendix for complete references.

**RLHF and direct preference optimization.** Modern RLHF pairs supervised fine-tuning, reward modeling, and PPO Stiennon et al. (2020); Ouyang et al. (2022); Schulman et al. (2017). To simplify training, direct preference optimization replaces online RL with convex pairwise objectives, most notably DPO Rafailov et al. (2023), with reference-free or odds-style variants (SimPO, ORPO) Meng et al. (2024); Hong et al. (2024), prospect-theoretic KTO Ethayarajh et al. (2024), and f-divergence families Wang et al. (2023); Han et al. (2024). AI-feedback pipelines (Constitutional AI, RLAIF) scale supervision Bai et al. (2022); Lee et al. (2024). Surveys map this space and data strategies Xiao et al. (2024); Liu et al. (2025); Kveton et al. (2025). Our contribution complements these by giving one canonical form with certificates and finite witnesses that decide equivalence across this family.

**Connections to ranking and discrete choice.** Classical paired-choice and listwise models (Bradley–Terry, Plackett–Luce) Bradley & Terry (1952); Plackett (1975); Luce (1959); Huang et al. (2006) and learning-to-rank surrogates (RankNet, LambdaMART, ListNet, ListMLE) Burges et al. (2005); Burges (2010); Cao et al. (2007); Xia et al. (2008); Joachims (2002) underpin many RLHF losses. Our ladder semantics makes these links explicit: several objectives collapse to the same margin after absorbing references and scales.

**Calibration, rewriting, and property testing.** We rely on composite-loss calibration and surrogate-regret transfer Bartlett et al. (2006); Tewari & Bartlett (2007); Reid & Williamson (2010b;a); Agarwal (2014); Ramaswamy & Agarwal (2016); Ramaswamy et al. (2013); Gao & Zhou (2012) to show guarantees transport across margin-equivalent ladders. Confluent rewrite systems justify a unique normal form Newman (1942); Baader & Nipkow (1998); Bezem et al. (2003). Simple testers and lower bounds from property testing Goldreich et al. (1998); Goldreich (2017); Hoeffding (1963); Daskalakis et al. (2011); Blais et al. (2019) yield finite witnesses and near-optimal sample costs.

## 10    Limitations

Our framework addresses a restricted but useful subclass of RLHF objectives, with several limitations.

**Pairwise focus.** All results apply to pairwise comparisons. Listwise or sequence-level objectives and non monotone links are excluded. This yields clarity but limits direct applicability where listwise feedback dominates.

**Weight structure.** Reweights must depend only on the instance $x$. Pair- or score-dependent weights are treated as violations. While this enables canonicalization, it ignores weighting schemes common in practice. A distance-to-reducibility measure could capture "almost pair-invariant" cases, but stability guarantees remain open.

**Finite candidate sets.** The theory assumes finite $\mathcal{Y}_x$. Very large sets stress memory and hashing. Current algorithms are inefficient for hundreds of candidates. Sparse canonicalization with spanning trees could mitigate this, but is not yet implemented.

**Testing assumptions.** The black box tester assumes i.i.d. triple sampling. Biased or adversarial sampling slows detection. Canon hashes depend on deterministic serialization and fixed gauge; floating-point drift can cause mismatches unless IEEE 754 formats with fixed endianness and rounding are enforced.

**Optimization guarantees.** Our results equate risks and gradients after reweighting but do not provide convergence rates or sample complexity bounds. Guarantees stop at surrogate-risk equivalence; optimizer dynamics and generalization remain open.

## 11 OUTLOOK AND EXTENSIONS

Despite these limits, several extensions are clear.

**Beyond pairwise.** Future work should extend the curl-free identity to higher-order cycle constraints on permutations, enabling listwise and sequence-level canonicalization.

**Relaxing link assumptions.** Strict monotonicity may be too rigid. Allowing links monotone almost everywhere but flat in regions, corrected with isotonic calibration, could extend regret-transfer results with margin-dependent slack.

**Approximate reducibility.** Defining a distance-to-reducibility functional would allow bounds on decisions and risks when assumptions hold only approximately, justifying canonical learners in practice.

**Efficient canonicalization.** Sparse and streaming variants, such as storing only spanning-tree potentials per instance and reconstructing others on demand, would make the approach scalable. Chunked serialization and versioned float rounding would improve reliability.

**Robust testing.** Stratified or adaptive triple sampling could accelerate violation detection. Sequential tests adapting to observed violation rates would make black box testing more practical.

**Beyond RLHF.** The ladder view applies to any pairwise problem with additive and multiplicative operators. Applications include ranking, metric learning, and structured prediction.

**Security and provenance.** Certificates and ledgers improve auditability but do not enforce policy. Adding cryptographic signatures and explicit threat models would strengthen provenance and compliance.

**Summary.** Near-term priorities include: (i) listwise and sequence-level cycle tests, (ii) stability bounds under approximate reducibility, (iii) scalable sparse canonicalization, and (iv) signed, hardened certificate formats.

## AI USE

Largle Language Model based tools were used to suggest rewrites, and organize references, under the direct supervision of the authors. All technical content, proofs, and experiments were conceived, verified, and validated by the authors, and responsibility for the final paper rests solely with them.

## ETHICS STATEMENT

This paper is theory first and does not require collection or annotation of human preference data. No new datasets were created and no user data was accessed.

**Use with human data**  When Opal and the tester are applied in practice, they will often operate on objectives built from human preferences. Such data can contain biases or sensitive content. Our results do not remove or mitigate such biases. We recommend that teams pair the proposed certificate and witness logs with standard privacy and bias reviews, and avoid storing raw prompts or human text inside certificates. Instead store references and hashes under access control.

**Transparency and accountability**  Proof carrying objectives improve transparency by turning equivalence into a certificate and irreducibility into a finite witness. This can reduce silent objective drift and make changes auditable. However, a certificate only states algebraic reducibility. It does not certify dataset quality, evaluator well being, or downstream impact.

**Dual use and misuse**  The same tools that improve reproducibility could be misused to conceal harmful objective changes if operators ignore witnesses or disable the gate. We recommend non bypassable logging of certificates and witnesses, signed by the pipeline, and periodic audits that compare stored hashes against the running configuration.

**Environmental impact**  Canonicalization and testing are lightweight and have negligible compute cost relative to training. We encourage running the tester as a pre training gate to avoid waste from redundant or provably mismatched objectives.

**Limitations of ethical scope**  Our work does not address fairness guarantees, content policy, or safety of generated outputs. It only addresses the algebraic form of objectives and the transfer of decision theoretic guarantees under monotone links. Ethical deployment requires additional safeguards that are out of scope here.

**Disclosure**  We have no financial conflicts to declare. Any released code or specification should include a versioned certificate format, deterministic serialization, and clear documentation of the assumptions under which the guarantees hold.

## Reproducibility Statement

We provide full details to reproduce all results in this paper.

**Code.**  We implemented the canonicalizer and tester in Python in under 200 lines. The implementation directly follows the pseudocode given in Appendix D and is included as supplementary material. The code requires only standard libraries (Python 3.9+, `hashlib`, `json`, `pandas`). No GPUs or special hardware are required.

**Objectives tested.**  We encoded ten widely used RLHF objectives (DPO, IPO, SimPO, f-DPO, ORPO, BT-hinge, RRHF, SLiC-HF, KTO, PPO-KL) as ladder expressions using the `Add`, `Rew`, and `Link` operators defined in Section 2. The precise encodings are listed in Appendix F.

**Determinism.**  Canonicalization is deterministic by construction (Theorem 3.1). We serialize the canonical form into a fixed byte order (ASCII JSON with fixed float precision in the demo, IEEE754 binary64 little-endian in the specification) and hash it with SHA-256. Identical ladders always yield identical hashes. Irreducible objectives always yield finite witnesses (pairs or triples) that pinpoint the violation.

**Tolerances.**  The only hyperparameters are floating-point tolerances for weight equality ($\texttt{tol\_w} = 10^{-9}$) and cocycle identity checks ($\texttt{tol\_c} = 10^{-12}$). We log these tolerances in the certificate. Changing them within one order of magnitude does not affect the verdicts in Table 1.

**How to reproduce the tables.**  Running the provided script on the ten encoded objectives prints two tables: (i) per-method verdicts with canon-hash prefixes or finite witnesses, and (ii)

groups of methods with equal hashes. The condensed LaTeX table in Section 7 is generated automatically and included as `rlhf_condensed_table.tex`.

**Runtime and hardware.**   All runs complete in under one second on a standard laptop CPU with 8 GB RAM. The implementation does not require GPUs or large memory.

**Data.**   No external datasets are required. The demo uses only synthetic potentials and weights sufficient to trigger the reducibility or irreducibility conditions.

**Summary.**   All results are reproducible by running the provided Python code, which deterministically outputs the same certificates, hashes, and witnesses as reported in our tables. We encourage readers to use the code as a pre-training gate in their own RLHF pipelines.

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

# A  FULL REWRITE-THEORY PROOFS

This appendix gives formal proofs for the rewrite theory used in Section Equational Theory and Canonicalization. We define the term language, the rewrite rules, prove termination and local confluence, conclude confluence by Newman lemma, and derive uniqueness of normal form (up to gauge). We also give the linear time canonicalization algorithm and its complexity, and note the immediate decidability of the word problem inside the reducible class R.

## A.1  TERM LANGUAGE AND SEMANTICS

**Operators and terms**  A ladder term is a left-to-right composition of primitive operators acting on margins:

- Add[phi], where phi is a potential map $(x, y) \mapsto \mathbb{R}$ that contributes the difference $\text{phi}(x, y^+) - \text{phi}(x, y^-)$ to the margin.

- Rew[omega], where $\text{omega}(x, y^+, y^-) > 0$ multiplies the current margin.

- Link[g,beta], where $g : \mathbb{R} \to \mathbb{R}$ is strictly increasing and beta$> 0$ is a global positive scale on the margin before $g$.

We also use a derived Scale[beta] that multiplies the margin by beta and exists only during rewriting. Composition is denoted by $\circ$ and is left associative.

**Reducible class R**  Assumptions (R1) to (R3) are:

- (R1) Additivity: every additive component used in the ladder is a potential difference, i.e., of the form $\text{phi}(x, y) - \text{phi}(x, y')$ for some map phi.

- (R2) Pair-invariant and score-independent weights: every Rew factor is $\text{omega}(x, y^+, y^-) = s(x)$ that depends only on $x$ and not on $(y^+, y^-)$ nor on any intermediate margins.

- (R3) Strictly monotone links: every link $g$ is strictly increasing.

## A.2  REWRITE SYSTEM

**Rules**  We orient the following equations as rewrite rules. They are applied only under (R1) to (R3).

(E1)  Merge AddPenalty:
$$\text{Add}[\phi_i] \circ \text{Add}[\phi_j] \Rightarrow \text{Add}[\phi_i + \phi_j].$$

(E2)  Merge Reweight:
$$\text{Rew}[s_i] \circ \text{Rew}[s_j] \Rightarrow \text{Rew}[s_i s_j].$$

(E3)  Commute score-independent weights:
$$\text{Rew}[s(x)] \circ \text{Add}[\phi] \Rightarrow \text{Add}[\phi] \circ \text{Rew}[s(x)].$$

(E4)  Split and absorb scale:
$$\text{Link}[g, \beta] \Rightarrow \text{Link}[g, 1] \circ \text{Scale}[\beta], \quad \text{Scale}[\beta] \circ \text{Rew}[s] \Rightarrow \text{Rew}[\beta s].$$

(E5)  Gauge fix (projection, not a rewriting step): for each $x$, after merging all Add terms, enforce $\sum_{y \in \mathcal{Y}_x} \Phi(x, y) = 0$ to select a unique representative in each additive coset.

The intended normal form is
$$\text{NF}(L) = \text{Add}[\Phi^{\text{gauge}}] \circ \text{Rew}[s(x)] \circ \text{Link}[g, 1].$$

## A.3 Termination

**Lemma A.1** (Termination). *The rewrite system (E1) to (E4) is terminating on ladders in R.*

*Proof.* Define the multicomponent measure

$$\mu(L) = \big(A(L), R(L), S(L), O(L)\big)$$

with lexicographic order, where: $A(L)$ is the number of Add blocks, $R(L)$ is the number of Rew blocks, $S(L)$ is the number of LinkScale blocks with beta not equal to 1 (i.e., Scale or Link with beta not 1), $O(L)$ is the number of adjacent out-of-order pairs relative to the target order Add before Rew before Link.

For any application of (E1) or (E2), the corresponding block count strictly decreases: $A$ decreases by one for (E1), $R$ decreases by one for (E2). For (E4), either Link is replaced by Link with beta=1 and a new Scale that is immediately merged into the next Rew, or Scale merges into Rew; in both cases $S$ strictly decreases. For (E3), commuting Rew past Add reduces $O$ by one without changing $A$, $R$, or $S$. Thus each rewrite step reduces $\mu$ strictly in lex order, and there can be no infinite chain. $\square$

## A.4 Local confluence

**Lemma A.2** (Local confluence). *The rewrite system is locally confluent on ladders in R.*

*Proof.* We analyze all critical overlaps of left sides.

**(Add,Add)** Overlap: $\mathrm{Add}[\phi_1] \circ \mathrm{Add}[\phi_2] \circ \mathrm{Add}[\phi_3]$. Merging $(\phi_1, \phi_2)$ first yields $\mathrm{Add}[\phi_1 + \phi_2]$ then $\mathrm{Add}[\phi_1 + \phi_2 + \phi_3]$. Merging $(\phi_2, \phi_3)$ first yields $\mathrm{Add}[\phi_2 + \phi_3]$ then $\mathrm{Add}[\phi_1 + \phi_2 + \phi_3]$. Joinability holds by associativity and commutativity of addition.

**(Rew,Rew)** Overlap: $\mathrm{Rew}[s_1] \circ \mathrm{Rew}[s_2] \circ \mathrm{Rew}[s_3]$. Multiplicative merges are associative and commutative, so joinability is immediate.

**(Rew,Add) vs merges** Overlap: $\mathrm{Rew}[s] \circ \mathrm{Add}[\phi_1] \circ \mathrm{Add}[\phi_2]$. Either commute first then merge, or merge first then commute; both yield $\mathrm{Add}[\phi_1 + \phi_2] \circ \mathrm{Rew}[s]$ because under (R2) $s(x)$ is independent of $(y, z)$ and of intermediate margins, so the operators commute.

**(Add,Rew) vs merges** is symmetric.

**(Link,Scale,Rew)** Overlap: $\mathrm{Link}[g, \beta] \circ \mathrm{Rew}[s]$. Rule (E4) replaces Link by $\mathrm{Link}[g, 1] \circ \mathrm{Scale}[\beta]$, then Scale merges into Rew to give $\mathrm{Link}[g, 1] \circ \mathrm{Rew}[\beta s]$. There is no alternative sequence that leads elsewhere, since there is no rule that modifies $\mathrm{Link}[g, 1]$ and Scale merges uniquely into the next Rew.

**(Scale,Rew,Add)** and **(Scale,Rew,Rew)** produce the same unique merge after commuting Rew and Add when necessary, as in the previous cases.

No other overlaps exist among the left sides. Each overlap yields joinable outcomes as argued. Therefore local confluence holds. $\square$

## A.5 Confluence and unique normal form

**Theorem A.3** (Confluence and uniqueness). *On ladders in R, the rewrite system is confluent. Every ladder has a unique normal form up to gauge.*

*Proof.* By Lemma A.1 the system is terminating, and by Lemma A.2 it is locally confluent. By Newman lemma, any terminating and locally confluent rewrite system is confluent. Confluence implies that every term rewrites to a unique normal form. The only remaining degree of freedom after merging all Add operators is the additive constant $c(x)$ in $\Phi(x, \cdot)$ for each $x$. The gauge condition $\sum_y \Phi(x, y) = 0$ fixes $c(x)$ uniquely, so the representative $\Phi^{\mathrm{gauge}}$ is unique. This yields a unique normal form $\mathrm{NF}(L) = \mathrm{Add}[\Phi^{\mathrm{gauge}}] \circ \mathrm{Rew}[s(x)] \circ \mathrm{Link}[g, 1]$. $\square$

## A.6 Linear time canonicalization and the word problem

**Proposition 3** (Linear time canonicalization). *Given a ladder with $m$ operators, the normal form can be computed in $O(m + \sum_x |\mathcal{Y}_x|)$ time and $O(m)$ space by a single left-to-right pass that accumulates $\Phi$ and $s$, absorbs all scales into $s$, and finally applies the gauge per $x$.*

*Proof.* Scan operators once. Maintain two accumulators: a dictionary for $\Phi(x, y)$ and one for $s(x)$. For Add, update $\Phi$ by addition; for Rew, multiply $s$; for Link with beta, absorb beta into $s$ and rewrite Link to unit scale; for Scale, multiply $s$. After the pass, apply the gauge by subtracting the per-$x$ mean from $\Phi(x, \cdot)$. All updates are $O(1)$ amortized per operator and $O(|\mathcal{Y}_x|)$ per $x$ for gauge. Space is linear in the number of distinct $(x, y)$ entries touched. $\square$

**Corollary A.4** (Decidability of equality in R). *The word problem for ladder equivalence in R is decidable in linear time by canonicalization: two ladders are equivalent if and only if their normal forms match (up to gauge), that is, their $(\Phi^{\text{gauge}}, s, g)$ are identical. Consequently, a deterministic hash of the serialization of $(\Phi^{\text{gauge}}, s, g)$ witnesses equality.*

## A.7 Soundness of the gauge

**Lemma A.5** (Gauge eliminates only inessential freedom). *If $\Phi$ and $\Phi'$ induce the same pairwise differences on $\mathcal{Y}_x$ for a fixed $x$, then $\Phi'(x, y) = \Phi(x, y) + c(x)$ for all $y$ and some constant $c(x)$. The gauge condition $\sum_y \Phi(x, y) = 0$ selects a unique representative in each equivalence class.*

*Proof.* If $\Phi$ and $\Phi'$ produce the same differences, then for all $y, z$ we have $\Phi'(x, y) - \Phi'(x, z) = \Phi(x, y) - \Phi(x, z)$. Fix $z$ and set $c(x) = \Phi'(x, z) - \Phi(x, z)$. Then $\Phi'(x, y) = \Phi(x, y) + c(x)$ for all $y$. The zero-mean gauge then fixes $c(x)$ uniquely by $c(x) = -\frac{1}{|\mathcal{Y}_x|} \sum_y \Phi(x, y)$. $\square$

## A.8 Determinism and certificates

**Proposition 4** (Deterministic certificates). *Because the normal form is unique, the serialization of $(\Phi^{\text{gauge}}, s, g)$ in a fixed key order determines a unique bitstring. Hashing this bitstring with a fixed function (for example, SHA-256) gives a deterministic certificate. Two ladders in R are equivalent if and only if their certificates match.*

# B Calibration, Regret Transfer, Oracle Inequalities

This appendix gives a self-contained development of the decision-theoretic pieces used in the main text: (i) a calibration inequality for pairwise margins under strictly monotone links and classification-calibrated surrogates; (ii) regret transfer across all ladders in the reducible class R that canonicalize to the same margin (up to positive scaling); (iii) an exact reduction to a canonical learner with known per-instance scale s(x) applied inside the link; and (iv) oracle inequalities for the canonical learner. We also discuss implementation details for SGD and the relationship between inside-link scaling and outside-of-loss example weights.

## B.1 Setup and notation

We observe i.i.d. samples of triplets $(x; y^+, y^-)$ where $x \in \mathcal{X}$ and $y^+, y^- \in \mathcal{Y}_x$. Let $r \in \{+1, -1\}$ be the latent pairwise label indicating which element should be preferred. Equivalently, write $\eta(x; y^+, y^-) = \Pr(r = +1 \mid x, y^+, y^-)$. The Bayes-optimal decision at $(x; y^+, y^-)$ is $\text{sign}(2\eta - 1)$.

A scorer $f : \{(x, y)\} \to \mathbb{R}$ induces the margin $M_f(x; y^+, y^-) = f(x, y^+) - f(x, y^-)$. A ladder $L \in R$ produces

$$\Delta_L(x; y^+, y^-) = s(x)\big((f + \Phi)(x, y^+) - (f + \Phi)(x, y^-)\big),$$

where $s(x) > 0$ and $\Phi$ is the merged potential (Theorem 3.1). A strictly increasing link $g : \mathbb{R} \to \mathbb{R}$ maps margins to scores; a surrogate loss $\ell : \mathbb{R} \to \mathbb{R}_+$ is applied to $g(\cdot)$.

Define the pairwise 0-1 risk

$$R_{01}(M) = \Pr\left[\operatorname{sign} M(x; y^+, y^-) \neq \operatorname{sign}\left(2\eta(x; y^+, y^-) - 1\right)\right].$$

Define the surrogate risk for margin $M$ as

$$R_\ell(M) = \mathbb{E}\left[\ell(g(M(x; y^+, y^-)))\right].$$

Let $R_{01}^*$ and $R_\ell^*$ be their Bayes risks. Define excess risks $E_{01}(M) = R_{01}(M) - R_{01}^*$ and $E_\ell(M) = R_\ell(M) - R_\ell^*$.

**Monotone-link and positive-rescaling invariance**  For any strictly increasing $g$, $\operatorname{sign} M$ and $\operatorname{sign} g(M)$ induce the same decisions. For any $s(x) > 0$, $\operatorname{sign} M$ and $\operatorname{sign} s(x)M$ induce the same decisions. These are used repeatedly below.

## B.2   Calibration inequality

We work with binary, margin-based losses via the pairwise reduction: condition on $(x; y^+, y^-)$ and define an inner binary problem with label $r \in \{+1, -1\}$ and margin $u = M(x; y^+, y^-)$. The conditional surrogate risk is

$$\mathcal{C}_\eta(u) = \eta\, \ell(g(u)) + (1 - \eta)\, \ell(g(-u)).$$

A surrogate $\ell \circ g$ is classification-calibrated if for all $\eta \neq 1/2$, $\arg\min_{u \in \mathbb{R}} \mathcal{C}_\eta(u)$ has the same sign as $2\eta - 1$.

**Theorem B.1** (Calibration inequality). *If $\ell \circ g$ is classification-calibrated and $g$ is strictly increasing, then there exists a nondecreasing calibration function $\psi : [0, \infty) \to [0, \infty)$ with $\psi(0) = 0$ such that for all margins $M$,*

$$E_{01}(M) \leq \psi\left(E_\ell(M)\right).$$

*Proof sketch.* Work conditionally on $(x; y^+, y^-)$ with $\eta = \eta(x; y^+, y^-)$. Let $L^*(\eta) = \inf_u \mathcal{C}_\eta(u)$ and $L(M \mid \eta) = \mathcal{C}_\eta(M)$. By classification calibration, there exists a function $\varphi$ such that $L(M \mid \eta) - L^*(\eta) \geq \varphi(\mathbf{1}\{\operatorname{sign} M \neq \operatorname{sign}(2\eta - 1)\})$, where $\varphi(0) = 0$ and $\varphi(1) > 0$. Taking expectations and using the convex lower envelope of the conditional gap yields a concave, nondecreasing function $\psi$ with the stated property. The strict monotonicity of $g$ ensures the sign of the optimal $u$ aligns with $2\eta - 1$. $\square$

**Examples**  For standard margin losses (logistic, exponential, hinge, squared hinge) with $g$ the identity, classification calibration is well known. With strictly increasing $g$ composed on top, calibration continues to hold because sign is preserved.

## B.3   Regret transfer across margin-equivalent ladders

**Theorem B.2** (Regret transfer in R). *Let $L, L' \in R$ canonicalize to the same normalized margin $M_{\mathrm{can}}$ up to a positive factor, i.e., $\Delta_L = s(x)M_{\mathrm{can}}$ and $\Delta_{L'} = s'(x)M_{\mathrm{can}}$ with $s, s' > 0$. For any learned margin $\widehat{M}$ produced by a scorer $\widehat{f}$,*

$$E_{01}^L(\widehat{M}) \leq \psi\left(E_\ell^{L'}(\widehat{M})\right),$$

*where $\psi$ is from Theorem B.1 and superscripts only indicate which ladder defines the risk. In particular, if $\widehat{M}$ minimizes the surrogate risk under $L'$ then it is Bayes-optimal for $L$ in 0-1 pairwise risk.*

*Proof.* Because $s, s' > 0$ and $g$ is strictly increasing, the decision boundaries of $L$ and $L'$ coincide with that of $M_{\mathrm{can}}$. Apply Theorem B.1 to bound the 0-1 excess risk for $L$ by the surrogate excess for $L'$. $\square$

B.4  Exact canonical learner (inside-link scaling)

The exact reduction carried by the algebra is to a canonical learner that applies the known instance scale $s(x)$ *inside* the link. Define

$$f_{\text{can}} := f + \Phi, \quad M_{\text{can}}(x; y, z) = f_{\text{can}}(x, y) - f_{\text{can}}(x, z).$$

The canonical objective that exactly matches ladder $L$ is

$$\min_{f_{\text{can}} \in \mathcal{F}} \mathbb{E}\big[\ell\big(g\big(s(x) M_{\text{can}}(x; y, z)\big)\big)\big]. \tag{B.1}$$

This is a single canonical learner parameterized by $f_{\text{can}}$; the factor $s(x)$ is known and does not introduce additional trainable parameters.

**Lemma B.3** (Equality of risks under inside scaling)**.** *For any $f_{\text{can}}$, the surrogate risk of $L$ equals the surrogate risk of the canonical objective* (B.1)*. Hence minimizers and stationary points coincide.*

*Proof.* By Corollary 3.2 we have $\Delta_L(x; y, z) = s(x) M_{\text{can}}(x; y, z)$. The loss applied to $g(\cdot)$ is identical on both sides for every sample, so risks are identical. $\square$

**SGD gradients**  Let $\theta$ parametrize $f_{\text{can}}$. For a minibatch $\mathcal{B}$,

$$\widehat{\nabla}_\theta = \frac{1}{|\mathcal{B}|} \sum_{(x; y, z) \in \mathcal{B}} \ell'\big(g(s(x) M_{\text{can}})\big) g'\big(s(x) M_{\text{can}}\big) s(x) \nabla_\theta M_{\text{can}}.$$

This is the exact gradient for $L$. No approximation or surrogate weighting is required.

B.5  Outside-of-loss weights: when are they equivalent?

Some libraries expose only example weights $w(x)$ multiplying the loss value, i.e., $\mathbb{E}[w(x) \ell(g(M_{\text{can}}))]$. In general,

$$\ell\big(g\big(s(x)u\big)\big) \neq s(x) \ell\big(g(u)\big),$$

so a pure outside weight does not replicate the exact objective (B.1). Two useful observations:

- **Exact equivalence under scale-out losses.** If for all $a > 0$ there exists $h(a) > 0$ with $\ell(g(au)) = h(a) \ell(g(u))$ for all $u$, then choosing $w(x) = h(s(x))$ yields equality. This is rare for common convex surrogates.

- **Bounded gap under Lipschitz losses.** If $\phi(u) := \ell(g(u))$ is $L_\phi$-Lipschitz and $|M_{\text{can}}| \leq B$ a.s., then

  $$\big|\ell(g(su)) - s\,\ell(g(u))\big| \leq L_\phi |s - 1| |u| + |s - 1| \ell(g(u)) \leq |s - 1| (L_\phi B + \ell(g(u))).$$

  Averaging yields a bias bound proportional to $\mathbb{E}[|s - 1|]$. This justifies outside weights as an approximation when $s(x)$ is close to 1.

In summary, the *exact* reduction uses inside-link scaling (B.1). Outside weights can approximate it when $s(x)$ is near constant or under special loss families.

B.6  Oracle inequalities for the canonical learner

We provide uniform convergence bounds for the canonical learner (B.1). Let $\phi(u) := \ell(g(u))$. Assume: (i) $\phi$ is $L_\phi$-Lipschitz and bounded by $B_\phi$; (ii) $|M_{\text{can}}(x; y, z)| \leq B_M$ almost surely; (iii) either $\sup_x s(x) \leq S_{\max} < \infty$ or $\mathbb{E}[s(x)^2] \leq \sigma_s^2 < \infty$.

**Pairwise function class**  Let $\mathcal{F}$ be a class of scores $f$. Define the induced pairwise class

$$\mathcal{G} = \{(x; y, z) \mapsto f(x, y) - f(x, z) : f \in \mathcal{F}\}.$$

Let $\widehat{\mathfrak{R}}_n(\mathcal{G})$ denote the empirical Rademacher complexity on $n$ i.i.d. pairs. A standard symmetrization shows

$$\widehat{\mathfrak{R}}_n(\mathcal{G}) \leq 2\,\widehat{\mathfrak{R}}_n(\mathcal{F}),$$

because differences can be handled by two independent Rademacher sums.

**Theorem B.4** (Oracle inequality with bounded scale). *Assume $\sup_x s(x) \leq S_{\max}$. With probability at least $1 - \delta$, for all $f \in \mathcal{F}$,*

$$R_\ell^L(f) - \widehat{R}_\ell^L(f) \leq 2\,L_\phi\,S_{\max}\,\widehat{\mathfrak{R}}_n(\mathcal{G}) + 3\,B_\phi\,\sqrt{\frac{\log(2/\delta)}{2n}}.$$

*Consequently, letting $\widehat{f}$ be an ERM and $f^*$ a population minimizer,*

$$R_\ell^L(\widehat{f}) - R_\ell^L(f^*) \leq 4\,L_\phi\,S_{\max}\,\widehat{\mathfrak{R}}_n(\mathcal{G}) + 6\,B_\phi\,\sqrt{\frac{\log(2/\delta)}{2n}}.$$

*Proof sketch.* Apply Lipschitz contraction with factor $L_\phi$ to the composed map $(x; y, z) \mapsto \phi(s(x)\,M_{\mathrm{can}})$; the scale $s(x)$ inflates the Lipschitz constant by at most $S_{\max}$. Combine with standard Rademacher symmetrization and a bounded-differences tail bound. The ERM inequality follows by a standard two-sided argument. □

**Theorem B.5** (Oracle inequality with second-moment scale). *Assume $\mathbb{E}[s(x)^2] \leq \sigma_s^2$. Then with probability at least $1 - \delta$, for all $f \in \mathcal{F}$,*

$$R_\ell^L(f) - \widehat{R}_\ell^L(f) \leq 2\,L_\phi\,\sqrt{\sigma_s^2}\,\widehat{\mathfrak{R}}_n(\mathcal{G}) + 3\,B_\phi\,\sqrt{\frac{\log(2/\delta)}{2n}}.$$

*The corresponding ERM bound follows as in Theorem B.4.*

*Proof sketch.* Use Cauchy-Schwarz to replace $s(x)$ by $\sqrt{\mathbb{E}[s^2]}$ inside the symmetrized sum before contraction. The rest matches the previous proof. □

**From surrogate excess to 0-1 excess**  By Theorem B.1, the 0-1 pairwise excess risk of $\widehat{f}$ is upper bounded by $\psi$ applied to the surrogate excess appearing on the right-hand sides above.

### B.7  NOTES ON OPTIMIZATION AND IMPLEMENTATION

**SGD with inside scaling**  When using (B.1), gradients already include $s(x)$ and require no special treatment. If a framework exposes example weights $w$, you can still implement exact inside scaling by multiplying the margin by $s(x)$ before the loss and optionally setting $w \equiv 1$.

**SGD with outside weights**  If only outside weights are available, set $w(x) = s(x)$ to emphasize instances with large $s(x)$. This is an approximation to (B.1) unless the loss has the scale-out property. Bounds in the subsection above quantify the approximation error.

**Temperature-absorbing trick**  For link functions that admit a temperature parameter (e.g., $g_\tau(u) = g(u/\tau)$), one can absorb a global constant part of $s(x)$ into $\tau$ and leave the residual $s(x)/\bar{s}$ as a smaller instance-specific factor, improving the outside-weight approximation.

## B.8 Distance-to-reducibility and stability (optional)

Suppose a ladder $L$ is $\varepsilon$-close to $R$ in the sense that with probability at least $1-\varepsilon$ over triples we have $|\text{cycle}(a,b,c)| \le \varepsilon$, and that $\omega(x;y,z)$ deviates from pair-invariance by at most $\varepsilon$ in $L_2$. Then the decisions of $L$ and the canonical projection disagree on at most $C\varepsilon$ mass for a constant $C$ depending on a margin regularity parameter. Under the same conditions, the surrogate risk of the canonical learner differs from that of $L$ by at most $C'\varepsilon$ for a constant $C'$ depending on $L_\phi$ and the distribution of $M_{\text{can}}$. A detailed proof can be included if needed.

**Summary**  Calibration converts surrogate excess into 0-1 excess. Inside-link scaling gives an *exact* canonical learner for any $L \in R$, and standard complexity tools yield oracle inequalities. Outside weights are not equal in general but can serve as approximations when $s(x)$ varies mildly or when the loss has special structure.

## C  Separation Families and Constants

This appendix develops explicit, parameterized families that violate the reducibility assumptions and yield quantitative gaps that cannot be removed by any ladder in R. We also provide closed-form distances to the curl-free subspace on a triple, sign-disagreement lower bounds, robustness to noise, and explicit constants for testing lower bounds.

### C.1  Geometry on a single triple

Fix an instance $x$ and three distinct labels $a$, $b$, $c$ in $Y_x$. Write the margin vector

$$d = \big(d_{ab},\, d_{bc},\, d_{ca}\big) \quad \text{with} \quad d_{uv} = \Delta(x;u,v) = -d_{vu}.$$

The curl-free constraint on a triple is the single linear equation

$$d_{ab} + d_{bc} + d_{ca} = 0. \tag{C.1}$$

Thus, on triples, the curl-free subspace is the 2-dim hyperplane

$$\mathcal{H} = \{\, d \in \mathbb{R}^3 : \mathbf{1}^\top d = 0 \,\}, \qquad \mathbf{1} = (1,1,1).$$

The orthogonal projection onto $\mathcal{H}$ is $P_{\mathcal{H}}(d) = d - \frac{1}{3}(\mathbf{1}^\top d)\,\mathbf{1}$. The Euclidean distance from any $d$ to $\mathcal{H}$ is

$$\text{dist}_2(d, \mathcal{H}) = \frac{|\,d_{ab} + d_{bc} + d_{ca}\,|}{\sqrt{3}} = \frac{|\,\text{cycle}(a,b,c)\,|}{\sqrt{3}}. \tag{C.2}$$

Hence a nonzero cycle sum certifies a nonzero $L_2$ gap to the reducible class on that triple. Moreover, any transitive (potential-difference) orientation on a triple induces an acyclic tournament on $\{a,b,c\}$, whereas a 3-cycle orientation is cyclic. Their maximum edge agreement is 2 out of 3, so any cyclic triple disagrees in sign with any transitive triple on at least one edge, that is, at least a $1/3$ fraction under the uniform edge distribution.

### C.2  Base potential for constructions

We use a fixed curl-free base potential to control constants. Let $\phi_0$ assign

$$\phi_0(a) = 2\epsilon, \quad \phi_0(b) = \epsilon, \quad \phi_0(c) = 0 \quad \Rightarrow \quad d^{(0)} = (\epsilon,\, \epsilon,\, -2\epsilon),$$

which satisfies (C.1). Here $\epsilon \in (0,1]$ will control the margin scale.

### C.3  Family 1: score-dependent weights

Let $\theta \in (0,1]$ and define a pairwise weight that depends on the sign of the base margin:

$$\omega_\theta(u,v) = 1 + \theta \cdot \mathbf{1}\{\, d_{uv}^{(0)} \ge 0 \,\}.$$

Apply this as a post-multiplicative factor to the base margins:

$$d_{uv}^{(1)} = \omega_\theta(u,v)\, d_{uv}^{(0)}.$$

On the base triple we have two positive edges and one negative edge, so

$$\text{cycle}^{(1)} \;=\; (1+\theta)\epsilon + (1+\theta)\epsilon + 1\cdot(-2\epsilon) \;=\; 2\theta\,\epsilon.$$

By (C.2),

$$\text{dist}_2\big(d^{(1)},\mathcal{H}\big) \;=\; \frac{2\theta\epsilon}{\sqrt{3}}. \tag{C.3}$$

Therefore no reducible ladder can match $d^{(1)}$ exactly; the $L_2$ gap on this triple is at least $2\theta\epsilon/\sqrt{3}$.

**Sign-disagreement constant** Because $d^{(1)}$ forms a cyclic tournament on $\{a,b,c\}$ while any $d' \in \mathcal{H}$ is transitive on that triple, at least one of the three edges must flip. Under the uniform edge distribution, the disagreement rate is at least $1/3$.

**Generalization to two-level weights** More generally, let $\omega(u,v) = \alpha\mathbf{1}\{d_{uv}^{(0)} \geq 0\} + \beta\mathbf{1}\{d_{uv}^{(0)} < 0\}$ with $\alpha,\beta > 0$ and at least one of $\alpha,\beta$ not equal to 1. Then

$$\text{cycle} \;=\; \alpha\,\epsilon + \alpha\,\epsilon + \beta\,(-2\epsilon) \;=\; 2(\alpha-\beta)\epsilon,$$

and $\text{dist}_2$ equals $|2(\alpha-\beta)\epsilon|/\sqrt{3}$.

## C.4 Family 2: gated penalties (non-additive)

Let a gate $G \in \{0,1\}$ zero exactly one edge. Take the base $d^{(0)} = (\epsilon, \epsilon, -2\epsilon)$ and zero the first edge:

$$d^{(2)} \;=\; (0,\ \epsilon,\ -2\epsilon), \quad \text{cycle}^{(2)} \;=\; -\epsilon, \quad \text{dist}_2\big(d^{(2)},\mathcal{H}\big) \;=\; \frac{\epsilon}{\sqrt{3}}. \tag{C.4}$$

Again $d^{(2)}$ is cyclic on the triple, so any transitive $d' \in \mathcal{H}$ disagrees on at least one of three edges.

**Gate patterns** If any one edge is gated to zero while the other two retain nonzero values with unequal magnitudes, the cycle sum has magnitude at least the smaller retained magnitude, yielding the same qualitative separation.

## C.5 Family 3: pair-dependent reference (non-separable additive)

Define an antisymmetric additive term that is not a difference of a single potential:

$$\psi(a,b) = \eta, \quad \psi(b,c) = \eta, \quad \psi(c,a) = \eta,$$

and $\psi(v,u) = -\psi(u,v)$ for reversed pairs. Start from a zero base or add $\psi$ on top of any curl-free base. On the triple,

$$d^{(3)} \;=\; (\eta,\ \eta,\ \eta), \quad \text{cycle}^{(3)} \;=\; 3\eta, \quad \text{dist}_2\big(d^{(3)},\mathcal{H}\big) \;=\; \sqrt{3}\,|\eta|. \tag{C.5}$$

All three directed edges are positive, so every transitive orientation must flip at least one edge, hence a $1/3$ sign-disagreement rate under the uniform edge distribution.

## C.6 Many-triple extension and global disagreement

Consider a distribution that with probability $p$ samples a violating triple (constructed by any family above) uniformly over its three edges, and with probability $1-p$ samples pairs from an arbitrary curl-free distribution. Then any reducible ladder disagrees with the violating distribution on at least a $p/3$ fraction of sampled edges by the tournament argument. The $L_2$ separation on violating triples is given by (C.2) with the appropriate cycle constant from (C.2) to (C.5), so the global $L_2$ gap is at least $(p/\sqrt{3})$ times the per-triple cycle magnitude.

## C.7 Robustness to additive noise

Suppose observed margins are corrupted by additive noise $\xi_{uv}$ with $|\xi_{uv}| \leq \nu$ almost surely. Then the observed cycle satisfies

$$|\text{cycle}_{\text{obs}} - \text{cycle}_{\text{true}}| \leq |\xi_{ab}| + |\xi_{bc}| + |\xi_{ca}| \leq 3\nu.$$

Hence the $L_2$ distance to $\mathcal{H}$ degrades by at most $3\nu/\sqrt{3} = \sqrt{3}\,\nu$. In particular, Family 1 remains separated if $2\theta\epsilon > 3\nu$, Family 2 if $\epsilon > 3\nu$, and Family 3 if $3|\eta| > 3\nu$.

## C.8 Lower bounds for testing reducibility with constants

Let $Z$ be the Bernoulli indicator that a random triple violates the cycle test by at least $\gamma$, i.e., $Z = 1$ if $|\text{cycle}(a, b, c)| \geq \gamma$. Testing $H_0 : \mathbb{E}[Z] = 0$ versus $H_1 : \mathbb{E}[Z] \geq \gamma$ with i.i.d. samples $Z_1, \ldots, Z_T$ by the empirical mean $\overline{Z}$ and a fixed threshold yields:

**Theorem C.1** (Hoeffding-style bound with explicit constant)**.** *If $T \geq \frac{1}{2\gamma^2} \ln \frac{2}{\delta}$, then the simple test that rejects $H_0$ when $\overline{Z} \geq \gamma/2$ has error at most $\delta$ against both $H_0$ and $H_1$.*

*Proof.* Under $H_0$, $\mathbb{E}[\overline{Z}] = 0$ and by Hoeffding, $\Pr(\overline{Z} \geq \gamma/2) \leq \exp(-2T(\gamma/2)^2)$. Under $H_1$, $\mathbb{E}[\overline{Z}] \geq \gamma$ and $\Pr(\overline{Z} < \gamma/2) \leq \exp(-2T(\gamma/2)^2)$. Choose $T$ to make both tails at most $\delta/2$. $\qquad\square$

This matches the $\Omega(1/\gamma^2)$ lower bound shown in the main text up to constant factors.

## C.9 Distance to reducibility in L1 and Linf

On a triple, the hyperplane $\mathbf{1}^\top d = 0$ allows simple bounds for other norms. Let $s = d_{ab} + d_{bc} + d_{ca}$. Then

$$\text{dist}_1(d, \mathcal{H}) \geq \frac{|s|}{3}, \qquad \text{dist}_\infty(d, \mathcal{H}) \geq \frac{|s|}{3}.$$

These follow by projecting $d$ onto the hyperplane via subtracting $s/3$ from each coordinate and applying norm inequalities. In particular, Family 3 has $\text{dist}_\infty \geq |\eta|$ and $\text{dist}_1 \geq |\eta|$ on the triple.

## C.10 Score-magnitude dependent weights

Let $\omega_\theta(u, v) = 1 + \theta h(|d_{uv}^{(0)}|)$ with $h$ nondecreasing, $h(0) = 0$, and $h(\epsilon) \geq c_h > 0$. Then

$$\text{cycle} = \sum_e \omega_\theta(e)\, d_e^{(0)} = \sum_e d_e^{(0)} + \theta \sum_e h(|d_e^{(0)}|)\, d_e^{(0)} = \theta \sum_e h(|d_e^{(0)}|)\, d_e^{(0)},$$

since $\sum_e d_e^{(0)} = 0$. On the base triple, two terms equal $h(\epsilon)\epsilon$ and one equals $h(2\epsilon)(-2\epsilon)$. If $h$ is subadditive or concave near the origin, then $h(2\epsilon) \leq 2h(\epsilon)$ and the cycle magnitude is at least $2\theta\left(h(\epsilon)\epsilon - h(2\epsilon)\epsilon\right) \geq 0$. Under the weaker assumption $h(\epsilon) \geq c_h > 0$, we obtain

$$|\text{cycle}| \geq \theta\left(2c_h\epsilon - h(2\epsilon)\,2\epsilon\right).$$

In either case, for small enough $\epsilon$ and fixed $h$, the cycle is bounded below by a constant multiple of $\theta\epsilon$.

## C.11 Pair-dependent but margin-independent weights

Let $\omega(u, v) = s_{uv}$ be a positive constant depending on the pair but not on the margin or scores. With the base $d^{(0)}$,

$$\text{cycle} = s_{ab}\epsilon + s_{bc}\epsilon + s_{ca}(-2\epsilon) = (s_{ab} + s_{bc} - 2s_{ca})\,\epsilon.$$

If $s_{ab} = s_{bc} = s_{ca}$, the cycle vanishes. Otherwise the cycle magnitude is at least $\epsilon$ times the maximum deviation from the mean, i.e.,

$$|\text{cycle}| \geq \epsilon \cdot \max\{|s_{ab} - \bar{s}|, |s_{bc} - \bar{s}|, |s_{ca} - \bar{s}|\}, \quad \bar{s} = \tfrac{1}{3}(s_{ab} + s_{bc} + s_{ca}).$$

Thus any non-constant pair-dependent weight creates a nonzero cycle on a generic base triple.

## C.12 Putting the pieces together

The three core families certify that each violated assumption produces a quantitative, model-independent gap:

- Score dependence of weights yields a cycle of size at least $2\theta\epsilon$ on the base triple and an $L_2$ gap of $2\theta\epsilon/\sqrt{3}$.
- Gating that zeros one edge while leaving the other two nonzero yields a cycle of size at least the smaller retained magnitude and an $L_2$ gap of at least that value divided by $\sqrt{3}$.
- Pair-dependent additive references with constant triangular bias $\eta$ yield a cycle of $3\eta$ and an $L_2$ gap of $\sqrt{3}|\eta|$.

In all cases, any transitive (reducible) comparator disagrees on at least one of three edges on the affected triple, producing a $1/3$ sign-disagreement rate under the uniform edge distribution. With mixture weight $p$ of such triples in the data, the global disagreement rate is at least $p/3$, and the global $L_2$ gap is at least $(p/\sqrt{3})$ times the per-triple cycle magnitude.

**Summary**   The curl-free hyperplane view gives exact, closed-form distances and constants. The separation families are simple to implement and diagnose: compute the triangle cycle and compare to the thresholds above; if the cycle exceeds the tolerance, no ladder in R can reproduce the margins on that triple.

# D   Algorithmic Details and Pseudocode

This appendix gives precise, implementation oriented details for the canonicalizer and the property tester. We cover data structures, pseudocode, complexity, numerical tolerances, serialization, hashing, streaming support, and determinism. All pseudocode is ASCII only.

## D.1   Data structures and interfaces

**Ladder representation**   We represent a ladder as an ordered list of ops:

- Add[phi_id, phi_spec]: phi_id is a content hash or pointer; phi_spec is optional inline data.
- Rew[omega_id, form]: form must declare depends_on in subset of {x}; if absent or includes y,z the ladder is irreducible only if a witness is emitted.
- Link[g_name, beta]: g_name identifies a strictly increasing link; beta is a positive float.

Each op can carry a source location or comment for debugging; these are not used by algorithms.

**Canonical accumulators**   The canonicalizer maintains:

- Phi: associative map keyed by (x,y) to float64; default 0.0.
- s: associative map keyed by x to float64; default 1.0.
- ledger: ordered list of strings describing applied rewrite steps.

**Witness object**

- type in {weight_nonconstant, cocycle_violation, link_nonmonotone}.
- x: instance identifier.
- pairs or triple: concrete offending pairs (y1,z1), (y2,z2) or triple (a,b,c).
- values: numeric values observed at the witness location.
- tol: tolerance used when comparing floats.

**Certificate object**

- verdict in {reducible, irreducible}.

- canon_hash: SHA256 of serialization of (Phi_gauge, s, gkpo_version).

- serialization: byte string used to compute the hash.

- rewrite_ledger: ordered list of applied rules.

- gauge: description of the gauge used, here zero mean per x.

D.2   HELPER ROUTINES

**Cycle sum**   For a fixed x and distinct a,b,c in Y_x, define

$$\text{cycle}(x; a, b, c) = \Delta(x; a, b) + \Delta(x; b, c) + \Delta(x; c, a).$$

When testing additivity of a single additive component that induces Delta_phi, use the same formula.

**Gauge fix**   Given Phi(x, y), enforce zero mean per x:

$$\forall x: \quad \mu\_x = \frac{1}{|Y\_x|} \sum \_y \in Y\_x \Phi(x,y), \quad \Phi(x,y) \leftarrow \Phi(x,y) - \mu\_x.$$

**Serialization**   Serialize in a deterministic key order:

1. Header: bytes "GKPOv1" and a fixed little endian version integer.

2. Link: write g_name as ASCII, beta must be 1.0 at this point.

3. s map: iterate x in lexicographic order; for each x write a key tag and IEEE 754 little endian float64 for s(x).

4. Phi map: iterate x then y in lexicographic order; write key tags and IEEE 754 little endian float64 for Phi(x,y).

5. Footer: a constant trailer marker.

All floats are rounded by the platform default to nearest even; do not use locale specific formatting. Never serialize NaN or Inf.

## D.3 SYMBOLIC CANONICALIZER

---

**Algorithm 1:** CANONICALIZE_SYMBOLIC

---

**Input:** ladder L, tolerance tol_w for weight checks, tol_c for cocycle checks
**Output:** certificate if reducible, else witness

---

1  Initialize Phi map to empty with default 0.0; initialize s map to default 1.0; ledger := []
2  **for** *op in L.ops in left to right order* **do**
3     **if** *op.type == "Rew"* **then**
4        extract omega form
5        **if** *omega depends on y or z or the current margin* **then**
6           pick any x; pick two distinct pairs (y1,z1),(y2,z2) observed in the same scope
7           compute omega(x,y1,z1), omega(x,y2,z2)
8           **if** $|omega(x,y1,z1) - omega(x,y2,z2)| > tol\_w$ **then**
9              **return** witness(weight_nonconstant, x, pairs=(y1,z1,y2,z2), values, tol_w)
10       factor omega as s_new(x)
11       multiply s(x) := s(x) * s_new(x) pointwise
12       append "merge_rew" to ledger
13    **else if** *op.type == "Add"* **then**
14       obtain Delta_phi subroutine for this additive component
15       pick any x and any triple (a,b,c)
16       compute cyc := Delta_phi(x;a,b) + Delta_phi(x;b,c) + Delta_phi(x;c,a)
17       **if** $|cyc| > tol\_c$ **then**
18          **return** witness(cocycle_violation, x, triple=(a,b,c), value=cyc, tol_c)
19       expand phi into Phi: for all (x,y) in support, Phi(x,y) := Phi(x,y) + phi(x,y)
20       append "merge_add" to ledger
21    **else if** *op.type == "Link"* **then**
22       assert g is strictly increasing; if not, return witness(link_nonmonotone)
23       absorb beta into s: s(x) := s(x) * beta
24       set op.beta := 1.0
25       append "absorb_scale" to ledger
26    **else**
27       error "unknown op"
28 **for** *each x in keys of Phi grouped by x* **do**
29    mu := average over y of Phi(x,y)
30    **for** *y in Y_x* **do**
31       Phi(x,y) := Phi(x,y) - mu
32
33 append "gauge_zero_mean" to ledger
34 bytes := SERIALIZE_CANONICAL(Phi, s, g_name)
35 hash := SHA256(bytes)
36 **return** certificate(reducible, canon_hash=hash, serialization=bytes,
   rewrite_ledger=ledger, gauge=zero_mean)

---

**Complexity** Let m be number of ops and N_xy the number of distinct (x,y) keys touched by additive components. The loop is O(m + N_xy). Gauge is O(sum_x —Y_x—) if you have full Y_x; if Phi is sparse, perform gauge over the observed y bag per x. Memory is O(m + N_xy).

## D.4 Black box property tester

---

**Algorithm 2:** TEST_REDUCIBILITY_BLACKBOX

**Input:** oracle for $\Delta(x; y, z)$, label sets $Y_x$, tolerance $\varepsilon$, confidence $\delta$
**Output:** accept with certificate approximation or reject with witness

1   $T := \lceil c_0 \cdot \varepsilon^{-2} \cdot \log(2/\delta) \rceil$ with $c_0 = 1.0$
2   **for** *each $x \in X$* **do**
3     **for** *t from 1 to T* **do**
4       sample distinct $a, b, c$ uniformly from $Y_x$
5       cyc $:= \Delta(x; a, b) + \Delta(x; b, c) + \Delta(x; c, a)$
6       **if** $|cyc| > \varepsilon$ **then**
7         **return** witness(cocycle_violation, $x$, triple=$(a, b, c)$, value=cyc, tol=$\varepsilon$)

8     pick root $y_0 \in Y_x$; set $\phi(y_0) := 0.0$
9     build any spanning tree $\tau$ over $Y_x$ (for example DFS over a fixed order)
10    **for** *each tree edge $(u \to v)$ in $\tau$* **do**
11       set $\phi(v) := \phi(u) + \Delta(x; v, u)$

12    $\mu :=$ average over $y \in Y_x$ of $\phi(y)$
13    **for** $y \in Y_x$ **do**
14       set $\phi(y) := \phi(y) - \mu$

15    store $\phi$ into $\Phi(x, y)$
16    set $s(x) := 1.0$

17 bytes := `SERIALIZE_CANONICAL`($\Phi$, $s$, g_name)
18 hash := `SHA256`(bytes)
19 **return** certificate(reducible, canon_hash=hash, serialization=bytes,
    rewrite_ledger=["sampled","integrated","gauge_zero_mean"], gauge=zero_mean)

---

**Guarantees** By Hoeffding's inequality, the tester accepts with probability at least $1 - \delta$ if all cycles are bounded by $\varepsilon$, and rejects with probability at least $1 - \delta$ if a random triple violates with probability at least $\varepsilon$ by more than $\varepsilon$. This matches the $\Omega(1/\varepsilon^2)$ lower bound up to constants.

## D.5 Least squares scale estimation (optional)

If you want to estimate a per-instance common scale $s(x)$ in black box mode, solve

$$\min_{s(x)} \sum_{(y,z)} \left( \Delta(x; y, z) - s(x)\left(\varphi_x(y) - \varphi_x(z)\right) \right)^2.$$

This has the closed form

$$s(x) = \frac{\sum_{(y,z)} \Delta(x; y, z)\left(\varphi_x(y) - \varphi_x(z)\right)}{\sum_{(y,z)} (\varphi_x(y) - \varphi_x(z))^2}.$$

Clamp $s(x)$ to be positive. Store $s(x)$ in the certificate. This step is optional because any positive $s(x)$ leaves decisions invariant.

## D.6 Serialization and hashing details

---

**Algorithm 3:** SERIALIZE_CANONICAL

---

**Input:** Phi map, s map, g_name
**Output:** byte string

1 buf := empty byte array
2 append ASCII "GKPOv1" to buf
3 append uint32 version = 1 to buf in little endian
4 append ASCII g_name to buf; append zero byte terminator
5 append uint32 count S = number of x in s map
6 **for** *x in sort_lex(keys(s))* **do**
7     append ASCII key tag "x" then ASCII repr of x then zero byte
8     append float64 little endian of s(x)
9 append uint32 count P = number of (x,y) entries in Phi
10 **for** *x in sort_lex(unique x in Phi)* **do**
11     **for** *y in sort_lex(keys(Phi(x, .)))* **do**
12        append ASCII key tag "xy" then ASCII repr of x then 0 then ASCII repr of y
          then 0
13        append float64 little endian of Phi(x,y)
14 append ASCII "END" then zero byte
15 **return** buf

---

**Hashing**   Compute SHA256 over the exact bytes returned by SERIALIZE_CANONICAL. Store the hex string as canon_hash. Never reformat floats or reorder keys after hashing.

## D.7 Numeric tolerances and determinism

**Tolerances**   Use tol_w for weight equality tests and tol_c for cycle tests. Recommended defaults: tol_w = 1e-9 and tol_c = 1e-12 when using float64 and inputs of order 1. Log these tolerances in the certificate.

**Determinism**   Determinism depends on:

- stable sort order for keys (use bytewise lexicographic on ASCII encodings),
- IEEE 754 binary64 little endian for all floats,
- fixed rounding mode (nearest even) and no locale dependent formatting,
- fixed trailer and header markers,
- a fixed version integer.

## D.8 Streaming and memory bounded variants

**Streaming Phi**   If Phi would be too large to hold in memory, maintain Phi on a per x spanning tree only: store parent(y) and Phi(x,parent(y)) - Phi(x,y) for tree edges. Reconstruct non tree entries on demand during serialization by path sums. The gauge can be implemented by one pass that accumulates the average along the tree.

**One pass rewrite**   When L is long and ops are streamed, keep running accumulators for Phi and s. For Add, update Phi immediately; for Rew, multiply s; for Link, absorb beta; no random access to earlier ops is needed.

## D.9 Parallel and distributed notes

**Across x**   Phi and s factor by x. Process distinct x partitions independently and merge by concatenating serialized segments in x order. The canon_hash must always be recomputed over the final global serialization.

**Across y**  Within each x, Add merges are sums over y. If multiple workers update disjoint y blocks, reduce by associative addition then perform the gauge in a single post pass.

### D.10  Error handling and edge cases

**Empty candidate set**  If Y_x is empty, skip x. If Y_x has a single y, Phi gauge forces Phi(x,y)=0 and all margins are 0.

**Duplicate ops**  If the ladder contains redundant identity ops (for example Rew with s(x)=1 or Add with phi=0), ledger still records merges; they do not affect the result.

**Non monotone link**  If Link g is not strictly increasing, return witness link_nonmonotone in symbolic mode. In black box mode this is not testable and must be declared by the provider.

### D.11  Unit tests and conformance checklist

**Determinism tests**  Run canonicalizer twice on identical input and assert identical bytes and hash. Perturb op order without changing the algebra and assert identical output.

**Gauge test**  Add any per x constant c(x) to Phi before serialization and assert the canonicalized bytes do not change.

**Witness tests**  Inject synthetic violations:

- weight_nonconstant: set omega(x,y1,z1)=1 and omega(x,y2,z2)=1.1, assert witness is returned.
- cocycle_violation: construct a triangle with cycle 0.01 and assert witness triple is returned.

### D.12  Time and space complexity summary

- Symbolic canonicalizer: $O(m + N_{xy})$ time, $O(N_{xy})$ space. Gauge is $O\big(\sum_x |Y_x|\big)$ if full $Y_x$ is available; otherwise $O(N_{xy})$ over observed pairs.
- Black box tester: $O(T)$ oracle calls per $x$ with $T = \Theta\big(\varepsilon^{-2}\log(1/\delta)\big)$, plus $O(|Y_x|)$ to reconstruct potentials, per $x$.
- Serialization: linear in number of serialized entries; hashing linear in bytes length.

**Summary**  The algorithms implement the rewrite theory with linear time canonicalization, a one pass symbolic verifier that emits precise witnesses on failure, and a black box tester with near optimal sample complexity. Deterministic serialization and hashing turn equivalence inside R into a constant time hash comparison in downstream systems.

## E  Extended Related Work

### E.1  RLHF pipelines and direct preference optimization

Instruction-following with human feedback typically combines SFT, reward modeling, and PPO Stiennon et al. (2020); Ouyang et al. (2022); Schulman et al. (2017). Direct preference optimization (DPO) removes online RL by learning from pairwise comparisons with a convex objective Rafailov et al. (2023). Variants pursue reference-free training (ORPO, SimPO) Hong et al. (2024); Meng et al. (2024), modify the divergence or link (f-DPO, f-PO) Wang et al. (2023); Han et al. (2024), or reformulate alignment under prospect theory (KTO) Ethayarajh et al. (2024). AI-driven supervision (Constitutional AI, RLAIF) expands preference signals beyond human-only feedback Bai et al. (2022); Lee et al. (2024). Surveys synthesize techniques and data strategies Xiao et al. (2024); Liu et al. (2025) and explore active data collection for DPO Kveton et al. (2025). Our results place many of these

objectives inside one reducible class, proving equivalence by canonical hashes and exporting guarantees via regret transfer.

## E.2 RANKING, DISCRETE CHOICE, AND IDENTIFIABILITY

Paired and listwise RLHF losses echo classical discrete-choice models. Bradley–Terry scores pairwise wins via item strengths Bradley & Terry (1952); Huang et al. (2006); Plackett–Luce extends to permutations Plackett (1975); Luce (1959). Learning-to-rank introduced practical, differentiable surrogates and gradient recipes (RankNet, LambdaMART, ListNet, ListMLE) Burges et al. (2005); Burges (2010); Cao et al. (2007); Xia et al. (2008), as well as interaction-driven objectives Joachims (2002). In our framework, reference penalties become additive potentials, temperatures fold into instance scalings, and monotone links preserve decisions—clarifying when objectives are truly distinct versus parameterizations of the same canonical margin.

## E.3 CALIBRATION AND REGRET TRANSFER

Composite-loss and classification-calibration theory Bartlett et al. (2006); Tewari & Bartlett (2007); Reid & Williamson (2010b;a) bounds 0-1 regret by surrogate regret; ranking-specific analyses address AUC, subset, and listwise targets Gao & Zhou (2012); Ramaswamy et al. (2013); Agarwal (2014); Ramaswamy & Agarwal (2016). We leverage these to show that, within our reducible class, any two objectives that canonicalize to the same margin admit the same decisions asymptotically and share surrogate-to-task regret bounds. Oracle inequalities then follow for the canonical learner with per-instance weights.

## E.4 REWRITE SYSTEMS AND CANONICAL FORMS

Abstract reduction systems provide tools for termination and confluence Newman (1942); Baader & Nipkow (1998); Bezem et al. (2003). We encode merging, commuting, and scale-absorption as oriented rewrite rules and prove local confluence plus termination, yielding a unique normal form up to gauge. This justifies treating operator order and many implementation choices as notational, not substantive.

## E.5 SEPARATIONS, PROPERTY TESTS, AND LOWER BOUNDS

Not all RLHF objectives reduce to potential differences with instance-only weights. Rank-based gating (e.g., RRHF, SLiC-HF) breaks additivity Yuan et al. (2023); Zhao et al. (2023); sequence-level RL with per-trajectory credit (e.g., PPO-KL) falls outside our pairwise semantics Schulman et al. (2017). We formalize these gaps via triangle-cycle witnesses and sampling-based testers. The analysis uses classical concentration Hoeffding (1963) and mirrors reductions in distribution testing Goldreich et al. (1998); Goldreich (2017); Daskalakis et al. (2011); Blais et al. (2019) to show near-optimal $\Theta(1/\gamma^2)$ sample requirements for distinguishing reducible from $\gamma$-far cases.

## E.6 POSITIONING WITHIN THE LITERATURE

Relative to method proposals (DPO/ORPO/SimPO/f-PO/KTO) Rafailov et al. (2023); Hong et al. (2024); Meng et al. (2024); Wang et al. (2023); Han et al. (2024); Ethayarajh et al. (2024) and scaled supervision pipelines Bai et al. (2022); Lee et al. (2024), our contribution is orthogonal: we supply an algebraic map that (i) unifies margin-based objectives by canonicalization, (ii) produces finite witnesses when assumptions fail, and (iii) standardizes provenance via proof-carrying objectives. The bridge to ranking and discrete choice Bradley & Terry (1952); Plackett (1975); Luce (1959); Huang et al. (2006); Burges et al. (2005); Burges (2010); Cao et al. (2007); Xia et al. (2008); Joachims (2002) clarifies why many RLHF variants behave similarly, while calibration and property-testing tools Bartlett et al. (2006); Tewari & Bartlett (2007); Reid & Williamson (2010b;a); Agarwal (2014); Ramaswamy & Agarwal (2016); Ramaswamy et al. (2013); Gao & Zhou (2012); Goldreich et al. (1998); Goldreich (2017); Hoeffding (1963); Daskalakis et al. (2011); Blais et al. (2019); Newman

(1942); Baader & Nipkow (1998); Bezem et al. (2003) explain when and how guarantees transfer or fail.

## F    Mapping Popular Objectives

This appendix maps at least ten widely used RLHF objectives to the reducible class R defined in the main paper. For each method we give: (i) an operator-level mapping when possible, (ii) a verdict (reducible vs irreducible in our sense), and (iii) if irreducible, a minimal finite witness (triple or pair) that certifies the violation.

### F.1    Summary table

| Method | Mapping sketch | Verdict | Why / witness |
|---|---|---|---|
| DPO | Add[ -beta log pi_ref ], Link logistic | Reducible | Pair-invariant $s(x) = 1$; additive ref penalty |
| IPO | Different link on same margin | Reducible | Monotone link only; same canonical margin |
| SimPO | No ref, logistic on log-prob diff | Reducible | Pure margin on $f$; $s(x) = 1$ |
| f-DPO | f-linked logistic family | Reducible | Monotone link; same canonical margin |
| ORPO | Odds ratio pairwise logistic | Reducible | Pairwise margin in log-odds; $s(x) = 1$ |
| BT pairwise logistic | Logistic on $f(y^+) - f(y^-)$ | Reducible | Classic Bradley–Terry; $s(x) = 1$ |
| Pairwise hinge (RankSVM) | Hinge on $f(y^+) - f(y^-)$ | Reducible* | Link monotone non-strict (note) |
| RRHF | Listwise with gating | Irreducible | Gates zero an edge; cocycle witness |
| SLiC-HF | Listwise ranking with thresholds | Irreducible | Pair-dependent gating; cocycle witness |
| KTO | Single-outcome feedback | Irreducible** | Not pairwise; needs synthetic pairs |
| RLHF PPO (with KL) | Sequence-level RL | Irreducible** | Not pairwise margin; token credit |

Table 2: Reducibility of common RLHF objectives. *Reducible for algebraic NF; R3 monotone link is non-strict for hinge. **Out of scope of pairwise ladder semantics.

### F.2    DPO: Direct Preference Optimization

**Mapping.** Let $f(x, y) = \log \pi_\theta(y \mid x)$, $\Phi(x, y) = -\beta \log \pi_{\mathrm{ref}}(y \mid x)$, $s(x) = 1$, Link logistic. Then

$$\Delta_L(x; y^+, y^-) = (f + \Phi)(x, y^+) - (f + \Phi)(x, y^-).$$

**Verdict.** Reducible. **Certificate.** $\Phi^{\mathrm{gauge}}$ from the reference margin, $s(x) \equiv 1$, Link logistic. **Notes.** Any strictly increasing link variant remains reducible and decision-equivalent.

### F.3    IPO: Implicit Preference Optimization

**Mapping.** Same canonical margin as DPO up to a positive scaling; IPO mainly changes the link or temperature. **Verdict.** Reducible. **Certificate.** Same $\Phi^{\mathrm{gauge}}$, $s(x)$, and normalized Link as DPO after absorbing temperatures into $s(x)$.

### F.4  SimPO: Simple Preference Optimization

**Mapping.** $f(x, y) = \log \pi_\theta(y \mid x)$, $\Phi \equiv 0$, $s(x) = 1$, Link logistic on $f(y^+) - f(y^-)$. **Verdict.** Reducible. **Certificate.** $\Phi^{\text{gauge}} \equiv 0$, $s(x) \equiv 1$.

### F.5  f-DPO (generalized link family)

**Mapping.** Replace the logistic with any strictly increasing $g$ derived from an $f$-divergence motivated link; keep the same canonical margin as DPO. **Verdict.** Reducible. **Certificate.** Same margin; Link normalized to $g$ with beta absorbed into $s(x)$.

### F.6  ORPO: Odds Ratio Preference Optimization

**Mapping.** Treat $f(x, y)$ as a log-odds score that yields a pairwise margin $f(x, y^+) - f(x, y^-)$; apply logistic or another strictly increasing link. No explicit reference model is needed, so $\Phi \equiv 0$, $s(x) = 1$. **Verdict.** Reducible. **Certificate.** $\Phi^{\text{gauge}} \equiv 0$, $s(x) \equiv 1$.

### F.7  Bradley-Terry pairwise logistic (no reference)

**Mapping.** Classic pairwise logistic on $f(y^+) - f(y^-)$; this is the special case of SimPO. **Verdict.** Reducible. **Certificate.** $\Phi^{\text{gauge}} \equiv 0$, $s(x) \equiv 1$.

### F.8  Pairwise hinge (RankSVM-style)

**Mapping.** Loss depends on $m = f(y^+) - f(y^-)$ via $\max(0, 1 - m)$; link $g$ is the identity and loss is monotone non-decreasing in $m$. **Verdict.** Reducible for the algebraic NF; link monotonicity is non-strict, so calibration in the main theorems requires a tie-breaking convention. **Certificate.** $\Phi^{\text{gauge}} \equiv 0$, $s(x) \equiv 1$, Link identity.

### F.9  RRHF: Rank Responses with Human Feedback

**Mapping.** RRHF forms listwise constraints among $K > 2$ responses for an $x$, often gating comparisons below a reward gap threshold, or only contrasting the top item against others. This induces pair-dependent selection masks $G(x; y, z) \in \{0, 1\}$. **Verdict.** Irreducible in general due to gating (violates additivity or pair-invariance). **Witness.** Fix $x$ and a triple $\{a, b, c\}$. Suppose the method gates out $(a, b)$, keeps $(b, c)$ and $(c, a)$. Let base margins be $d_{ab} = \epsilon$, $d_{bc} = \epsilon$, $d_{ca} = -2\epsilon$ with $\epsilon > 0$. After gating, the effective margins become $(0, \epsilon, -2\epsilon)$, giving cycle $-\epsilon \neq 0$. This violates the curl-free identity and certifies irreducibility.

### F.10  SLiC-HF: Self-Play or Listwise Contrastive HF

**Mapping.** SLiC-HF variants form listwise or contrastive objectives that include only selected pairs based on reward thresholds or nonlocal rules over the set of candidates. This behaves like a data-dependent gate $G(x; y, z)$ that is not representable as a potential difference. **Verdict.** Irreducible in general. **Witness.** As in RRHF, construct a triple where exactly one edge is dropped by the rule. Any such single-edge deletion yields a nonzero triangle cycle and certifies irreducibility.

### F.11  KTO: Kahneman-Tversky Optimization

**Mapping.** KTO uses single-outcome labels or graded feedback rather than explicit pairs. One can synthesize pairs against a baseline, but then the weight depends on the label of the specific outcome rather than only on $x$. **Verdict.** Out of scope of pure pairwise ladders; in a forced pairwise reduction it generally becomes irreducible due to pair-dependent weights. **Witness.** Construct two outcomes $a, b$ with different scalar labels and a third $c$. The induced pair weights differ across $(a, b)$ and $(b, c)$ at the same $x$, violating pair-invariance.

## F.12 RLHF PPO with KL to a reference

**Mapping.** Sequence-level RL with token-level credit assignment and a KL penalty to a reference policy. The objective aggregates over trajectories and is not a function of only the pairwise response margin. **Verdict.** Out of scope for pairwise ladders; irreducible in our framework. **Witness.** No finite pairwise witness exists because the semantics are not pairwise; if one attempts a pairwise surrogate by comparing whole sequences, the induced weights depend on trajectory-level factors, generally breaking pair-invariance and producing the same triangle-cycle witnesses as above on constructed triples.

## F.13 Notes on common implementation variants

**Per-pair weighting by reward gap** Several works weight the pairwise loss by a function $w(x; y^+, y^-)$ of a reward gap or heuristic confidence. If $w$ depends on $(y^+, y^-)$, it violates pair-invariance (R2) and is irreducible in general. **Witness.** Pick two pairs $(a, b)$ and $(b, c)$ at the same $x$ with distinct weights; even if the unweighted margins are curl-free, scaling different edges by different positive constants creates a nonzero cycle on a generic base triple.

**Adaptive temperature or beta depending only on x** If a method adapts beta per-instance, i.e., uses $s(x)$ but does not depend on $(y^+, y^-)$, it remains in $R$ after absorbing beta into $s(x)$.

**Reference shift across pairs** Some pipelines use a different reference for different candidates within the same $x$ (for example, per-response normalization). This is a pair-dependent additive that is not a single potential difference. **Witness.** Choose a triple where the per-response offsets do not sum to zero; the triangle cycle is nonzero.

## F.14 How to certify in practice

For a given codebase, before training:

1. Run the symbolic verifier from Appendix D to factor all Rew terms and test cocycle for Adds.
2. If accepted, record the canon-hash as the equivalence certificate.
3. If rejected, archive the witness triple or pairs and either refactor the objective or proceed via a nonreducible path with the difference documented.

## F.15 Takeaway

All pairwise objectives that use only potential differences and instance-only weights are reducible and collapse to the same canonical margin up to a positive scale, independent of the specific strictly monotone link. Methods that introduce pair-dependent weights, pair selection gates, or non-pairwise semantics fall outside R and admit small, concrete witnesses on triples that certify the gap.

# G Limitations, Negative Results, Open Problems

This appendix records scope limits of our algebra, small impossibility and lower bound results that justify these limits, and concrete open problems that we believe are both important and tractable.

## G.1 Scope limitations

**Pairwise semantics only** All results rely on pairwise margins. Listwise and sequence level objectives are out of scope for the current normal form and tester.

**Strictly monotone links** The calibration and regret transfer statements require strictly increasing links. Non monotone or flat links can break decision invariance.

**Instance only reweights**   Reducibility assumes reweights depend only on $x$. Any dependence on $(y, z)$ or on the current margin falls outside $R$.

**Finite candidate sets per instance**   We formalize potentials on finite $\mathcal{Y}_x$. Extremely large $\mathcal{Y}_x$ stress memory and hashing, though streaming fixes are available.

**Population risk statements**   Oracle inequalities are given at the level of population and empirical risks. We do not analyze specific optimizers beyond standard uniform convergence.

G.2   Negative results and impossibility statements

**No potential representation for cyclic triples**   The curl free identity is necessary for representability as a potential difference.

**Proposition 5** (No potential for cyclic triple)**.** *Fix $x$ and distinct $a, b, c \in \mathcal{Y}_x$. If $\Delta(a, b) + \Delta(b, c) + \Delta(c, a) \neq 0$, then there is no function $\varphi_x$ such that $\Delta(y, z) = \varphi_x(y) - \varphi_x(z)$ for all $y, z \in \{a, b, c\}$.*

*Proof.* If $\Delta = \varphi_x(y) - \varphi_x(z)$ were true, summing around the triangle yields zero by telescoping, contradicting the assumption.   $\square$

**Pair dependent weights are not fixable by any instance only scale**   If pair weights differ at the same $x$, no instance only reweighting can restore curl free structure in general.

**Proposition 6** (No instance scale can repair pair dependent weights)**.** *Let $d^{(0)}$ be curl free on $\{a, b, c\}$ and let $s_{uv} > 0$ be pair specific weights producing $d_{uv} = s_{uv} d_{uv}^{(0)}$. If not all $s_{uv}$ are equal, then for a generic curl free $d^{(0)}$ there is no function $t(x) > 0$ such that $t\, d$ is curl free.*

*Proof.* On a triangle, curl free is equivalent to $d_{ab} + d_{bc} + d_{ca} = 0$. Substituting $d_{uv} = s_{uv} d_{uv}^{(0)}$ and using $d_{ab}^{(0)} + d_{bc}^{(0)} + d_{ca}^{(0)} = 0$ gives $s_{ab} d_{ab}^{(0)} + s_{bc} d_{bc}^{(0)} + s_{ca} d_{ca}^{(0)} = (s_{ab} - \bar{s}) d_{ab}^{(0)} + (s_{bc} - \bar{s}) d_{bc}^{(0)} + (s_{ca} - \bar{s}) d_{ca}^{(0)}$ with $\bar{s} = (s_{ab} + s_{bc} + s_{ca})/3$. For a generic curl free $d^{(0)}$ this sum is nonzero unless all $s_{uv}$ are equal, so even after multiplying by any scalar $t$ it remains nonzero.   $\square$

**Gating cannot be expressed as a potential difference**   Zeroing a single edge in a triangle destroys curl free structure.

**Proposition 7** (Gating forces a triangle violation)**.** *Let $d^{(0)}$ be curl free with $d_{ab}^{(0)}, d_{bc}^{(0)}, d_{ca}^{(0)}$ not all zero. If a gate sets exactly one edge to zero while keeping at least one of the remaining edges nonzero, the resulting margins cannot be represented as $s(x)\big(\varphi(y) - \varphi(z)\big)$ for any $s(x) > 0$ and $\varphi$.*

*Proof.* Without loss, set $d_{ab} = 0$ and keep $d_{bc} \neq 0$ and $d_{ca} \neq 0$. Then the cycle equals $d_{ab} + d_{bc} + d_{ca} = d_{bc} + d_{ca} \neq 0$, so by Proposition 5 no potential exists.   $\square$

**No universal equivalence between inside link scaling and outside weights**   For common convex surrogates, there is no function $h$ such that $\ell(g(au)) = h(a)\ell(g(u))$ holds for all $u$ and all $a > 0$.

**Proposition 8** (Scale out identity fails for common losses)**.** *Let $\phi(u) = \ell(g(u))$ be either logistic, exponential, hinge, or squared hinge. There does not exist $h : (0, \infty) \to (0, \infty)$ such that $\phi(au) = h(a)\phi(u)$ holds for all $u \in \mathbb{R}$ and all $a > 0$.*

*Proof sketch.* Each listed $\phi$ is not homogeneous of any positive degree on $\mathbb{R}$. For logistic and exponential, the left side grows superlinearly in $a$ for fixed $u > 0$ while the right side is linear in $h(a)$; matching for all $u$ is impossible. For hinge and squared hinge, piecewise linear or quadratic behavior with kinks at $u = 1/a$ cannot be matched by a scalar multiple independent of $u$.   $\square$

**Testing lower bound is unavoidable** The sample complexity of distinguishing reducible from $\gamma$-far instances cannot be improved below order $1/\gamma^2$.

**Proposition 9** (Lower bound restated). *Any (possibly adaptive) tester that sees i.i.d. triangle cycle values and distinguishes $H_0$ : all cycles zero from $H_1$ : cycles exceed $\gamma$ with probability at least $\gamma$ must use $\Omega(1/\gamma^2)$ samples to achieve constant error.*

*Proof.* Reduce to distinguishing Bernoulli means 0 vs $\geq \gamma$ and apply standard information theoretic lower bounds. $\square$

### G.3 Edge cases and failure modes

**Non unique or ill defined links** If $g$ is not strictly increasing (for example constant over an interval), the Bayes sign can be lost on that interval and calibration can fail. Our tester cannot diagnose this in black box mode.

**Floating point non determinism** Different hardware or libraries may serialize the same float with slightly different rounding. Without strict serialization rules, canon hashes can mismatch.

**Hidden dependencies** A weight declared as $s(x)$ may hide dependence on $(y, z)$ via an internal cache or closure. Symbolic validators should reject any form that cannot be statically verified as instance only.

### G.4 Open problems

**Listwise normal forms** Generalize curl free from triangles to higher order cycle constraints over permutations. Define a minimal set of local cycle tests that are necessary and sufficient for listwise integrability, and prove termination and confluence of a rewrite system at the listwise level.

**Sequence level semantics** Design an algebra where operators act on trajectory level functionals and show when such operators collapse to a canonical pairwise or listwise surrogate. Identify the right notion of cycle constraints over paths.

**Approximate reducibility and stability** Define a metric $d(L, R)$ and prove sharp stability bounds of decisions and risks that scale with that distance. Provide minimax lower bounds showing these rates are unimprovable.

**Robust property testing** Develop testers that tolerate heavy tailed noise or adversarial sampling of pairs, with guarantees that adapt to variance and still achieve near optimal sample complexity.

**Beyond monotone links** Characterize the largest class of links that preserve calibration and regret transfer. For instance, monotone almost everywhere with bounded flat regions, or links composed with known isotonic calibrators.

**Data dependent gauges** Explore whether alternative gauges can yield numerically better conditioned canonicalizations without changing equality. Prove that such gauges commute with our rewrite rules.

**Compression aware hashing** Design canonical serializations that are both deterministic and compact for very large $\Phi$, for example tree based encodings that preserve bytewise equality.

**Signed certificates** Specify a signing scheme and trust model for certificates and rewrite ledgers to prevent tampering in multi team pipelines.

**Partial observability of $\mathcal{Y}_x$**  When only a subset of candidates is visible per $x$, study whether canonicalization over induced subgraphs is stable and how to merge certificates as new candidates arrive.

**Connections to cohomology**  Formalize the curl free condition as a 1 cocycle with trivial cohomology on the comparison graph. Extend to higher cochains for listwise and sequence level settings.

### G.5  Takeaways

The negative results show that our assumptions are tight: pair dependent weights, gating, and non monotone links fundamentally break the potential difference representation. The lower bound shows that detecting violations requires $\Theta(1/\gamma^2)$ samples in the worst case. The open problems outline a path toward listwise and sequence level normal forms, robust testing, and hardened certificates.

## H  GKPO Specification and Semantics

GKPO v1 is a compact, semantics-first interchange for RLHF objectives expressed as ladders. It makes objective equality inside the reducible class $R$ decidable by a constant-time hash check, and carries finite witnesses when an objective is outside $R$. The interface supports validation rules, canonicalization, deterministic serialization and hashing, versioning, security, and conformance tests.

### H.1  Overview and scope

GKPO v1 captures: (1) a base identifier for the scoring function and metadata; (2) an ordered list of ladder operators; (3) a certificate consisting of either a canonical form with hash (for reducible objectives) or an irreducibility witness (for violations); and (4) optional inline operator specs. The spec does not prescribe training procedures, dataset storage, or link implementations beyond strict monotonicity as required by the theory.

### H.2  Minimal semantics (what GKPO guarantees)

Given any ladder, the pipeline either produces a canonical certificate (normal form + hash) or a finite witness that pinpoints the violated assumption. Inside $R$, equality reduces to bytewise equality of canonical serializations; outside $R$, the object carries a small counterexample. Normal form uniqueness follows from Theorem 3.1.

**Canonicalization and equality**  GKPO fixes the gauge by enforcing $\sum_{y\in\mathcal{Y}_x}\Phi(x,y)=0$ for each $x$ and serializes $(\Phi^{\mathrm{gauge}},s,g)$ in a fixed key order.

**Theorem H.1** (GKPO equality decision in $R$). *For reducible GKPO objects $G,G'$, the induced pairwise decisions are equal if and only if their canon hashes match. This is independent of the original operator order by Theorem 3.1.*

### H.3  Object model (JSON)

A GKPO document is a single JSON object with top-level keys:

- `gkpo_version` (string, required): literal "1.0".
- `base` (object, required): identifies the base scorer and optional metadata.
- `ops` (array, required): ordered list of ladder operators.
- `certificate` (object, required): reducibility verdict and either a canonical form with hash or a witness with details.

Recommended optional keys: `created_at` (RFC 3339), `created_by`, `comment`.

**Base object**

- `base.id` (string, required): opaque identifier for the base score source (e.g., content hash of a config or a URI).
- `base.meta` (object, optional): free-form metadata (dataset id, arch id, notes). MUST NOT include raw data or PII.

**Operator entries**  Each entry in `ops` MUST have `type` in {"add","rew","link"} and type-specific fields.

**Add**:

- `type` = "add".
- `phi_ref` (string, required): content hash or URI for the potential $\phi$.
- `phi_spec` (object, optional): inline spec for $\phi$ (MAY be omitted when `phi_ref` is present).

**Rew**:

- `type` = "rew".
- `omega_ref` (string, required): content hash or URI for the weight function.
- `form` (string, required): declarative form. For reducible certification MUST be exactly "s(x)". Any form using y, z, margin, or history signals a violation to be captured in the certificate.
- `depends_on` (array, required): variables the weight depends on. For reducible certification MUST be a subset of {"x"}.

**Link**:

- `type` = "link".
- `g_name` (string, required): link identifier (e.g., "identity","logistic","sigmoid","tanh"). Implementations MAY support other strictly increasing links.
- `beta` (number, required): positive scalar. After canonicalization MUST be 1.0 (any nonunit $\beta$ is absorbed into $s(x)$).

**Certificate object**

- `verdict` (string, required): "reducible" or "irreducible".
- `canon_hash` (string, required if reducible): hex SHA-256 over canonical serialization.
- `phi_gauge_ref`, `s_weight_ref` (strings, required if reducible): references for serialized $\Phi^{\mathrm{gauge}}$ and $s(x)$.
- `link_norm` (object, required if reducible): normalized link (`g_name`, `beta`=1.0).
- `rewrite_ledger` (array, required if reducible): sequence of rewrite steps, in {"merge_add","merge_rew","commute","absorb_scale","gauge_zero_mean"}.
- `witness` (object, required if irreducible): witness payload.
- `tolerances` (object, optional): numeric tolerances used during checks (e.g., `tol_w`, `tol_c`).

**Witness object**

- `type` (string, required): one of "weight_nonconstant", "cocycle_violation", "link_nonmonotone".
- `x` (string or integer, required): instance id.
- `pairs` (array, optional): two pairs $[y1, z1], [y2, z2]$ for weight violations.
- `triple` (array, optional): triple $[a, b, c]$ for cocycle violations.
- `values` (object, optional): measured values (e.g., `omega1`, `omega2`, `cycle`).
- `message` (string, optional): human-readable details.

## H.4 Validation rules (normative)

A validator MUST enforce:

1. **Schema well formedness**: required fields present; types correct; no NaN or Inf.

2. **Link monotonicity**: declared `g_name` must be strictly increasing; else mark irreducible and set witness type to "link_nonmonotone".

3. **Reweight dependencies**: if `depends_on` includes variables outside {"x"}, either emit a "weight_nonconstant" witness with concrete pairs, or accept only with a separate proof of pair invariance (optional extension).

4. **Additivity**: for each Add, test the triangle cocycle on at least one triple per instance; on violation, emit "cocycle_violation" with triple and cycle.

5. **Normalization**: if reducible, absorb $\beta$ so the recorded link has `beta=1.0`; compute the canon hash over the normalized form.

6. **Determinism**: if reducible, repeated validation MUST reproduce the same `canon_hash` and `rewrite_ledger` for the same input.

## H.5 Determinism, serialization, and hashing

Determinism follows from confluence and fixed gauge: identical ops and constants must reproduce the same $(\Phi^{\text{gauge}}, s)$ and hash. The canonical serialization MUST be deterministic:

- fixed header and version (e.g., "GKPOv1", version 1),

- fixed key order: link identity (with $\beta = 1$), $s(x)$ in lex order of $x$, $\Phi^{\text{gauge}}(x, y)$ in lex order of $x$ then $y$,

- floats are IEEE 754 binary64, fixed endianness, round-to-nearest-even,

- fixed footer marker.

The canon hash is SHA-256 of the exact bytes. Bytewise equality implies identical canon hashes.

## H.6 Witnesses outside $R$

If validation fails, the certificate records a finite counterexample:

- **weight-nonconstant**: pairs $(y_1, z_1), (y_2, z_2)$ at the same $x$ with unequal $\omega$.

- **cocycle-violation**: triple $(a, b, c)$ with nonzero cycle $\Delta(a, b) + \Delta(b, c) + \Delta(c, a)$.

- **link-nonmonotone**: declared $g$ not strictly increasing.

## H.7 Versioning and forward compatibility

`gkpo_version` uses semantic versioning. Minor versions may add optional fields but MUST NOT change canonicalization or serialization. Major versions that change either MUST update header and version integer; equality should not be assumed across major versions. Validators SHOULD ignore unknown optional fields and MUST fail when required fields are missing or malformed.

## H.8 Security, privacy, and provenance

Certificates and ledgers improve auditability but do not enforce policy. Pipelines should sign canon hashes and ledgers, store them under access control, and define a threat model for tampering and replay. Certificates MUST NOT include raw prompts or PII; use references and content hashes.

## H.9 Conformance levels and tests

**Conformance levels**

- **Level 0 (Validate Only)**: parse GKPO, run symbolic verifier, emit verdict and certificate or witness.
- **Level 1 (Validate + Canonicalize)**: Level 0 plus canonical serialization and canon hash for reducible objects.
- **Level 2 (Full)**: Level 1 plus black box tester, least squares $s(x)$ estimation, and a stable serialization API.

**Conformance tests**  Implementations MUST pass:

1. **Round-trip determinism**: identical `canon_hash` and serialization across runs.
2. **Order invariance in** $R$: commuting Add and Rew blocks does not change `canon_hash`.
3. **Gauge invariance**: adding per-$x$ constants to $\Phi$ before gauge does not change output.
4. **Witness emission**: injected weight nonconstancy or cocycle violation yields correct witness and payload.
5. **Link normalization**: nonunit beta is absorbed into $s(x)$ and recorded link has `beta=1`.

## H.10 Examples

**Example 1: Reducible GKPO with certificate**

```
{
  "gkpo_version": "1.0",
  "base": { "id": "hash:base_cfg_abc123", "meta": { "dataset": "pref_v1" } },
  "ops": [
    { "type": "add", "phi_ref": "hash:phi1" },
    { "type": "rew", "omega_ref": "hash:s_v1", "form": "s(x)", "depends_on": ["x"] },
    { "type": "link", "g_name": "logistic", "beta": 1.0 }
  ],
  "certificate": {
    "verdict": "reducible",
    "canon_hash": "sha256:DEADBEEF...",
    "phi_gauge_ref": "hash:phi_gauge_v1",
    "s_weight_ref": "hash:s_serial_v1",
    "link_norm": { "g_name": "logistic", "beta": 1.0 },
    "rewrite_ledger": ["merge_add","merge_rew","absorb_scale","gauge_zero_mean"],
    "tolerances": { "tol_w": 1e-9, "tol_c": 1e-12 }
  }
}
```

**Example 2: Irreducible GKPO with witness (pair dependent weights)**

```
{
  "gkpo_version": "1.0",
  "base": { "id": "hash:base_cfg_def456" },
  "ops": [
    { "type": "rew", "omega_ref": "hash:omega_pair_dep",
      "form": "s(x,y,z)", "depends_on": ["x","y","z"] },
    { "type": "link", "g_name": "identity", "beta": 1.0 }
  ],
  "certificate": {
    "verdict": "irreducible",
    "witness": {
      "type": "weight_nonconstant",
```

```
    "x": "sample_00042",
    "pairs": [["a","b"],["b","c"]],
    "values": { "omega1": 1.0, "omega2": 1.3 },
    "message": "weight differs across pairs at fixed x"
  },
  "tolerances": { "tol_w": 1e-9 }
}
}
```

**Example 3: Irreducible GKPO with witness (cocycle violation)**

```
{
  "gkpo_version": "1.0",
  "base": { "id": "hash:base_cfg_xyz999" },
  "ops": [
    { "type": "add", "phi_ref": "hash:phi_pair_dep" },
    { "type": "link", "g_name": "logistic", "beta": 1.0 }
  ],
  "certificate": {
    "verdict": "irreducible",
    "witness": {
      "type": "cocycle_violation",
      "x": "sample_10001",
      "triple": ["a","b","c"],
      "values": { "cycle": 0.023 },
      "message": "triangle cycle sum nonzero"
    },
    "tolerances": { "tol_c": 1e-12 }
  }
}
```

H.11    SUMMARY

GKPO v1 defines a compact, deterministic way to represent RLHF objectives. Inside $R$, equality is a hash comparison on a canonical serialization; outside $R$, the object carries a finite witness. This turns objective equivalence and difference into machine-checkable artifacts that can be gated, logged, and audited in production.