# OpenReview forum: "Opal: An Operator-Algebra View of RLHF Objectives"
_ICLR.cc/2026/Conference — Submitted to ICLR 2026_

### Official Review · Reviewer_ybHd · 2025-10-26

**Soundness:** 2
**Presentation:** 1
**Contribution:** 2
**Rating:** 0
**Confidence:** 4

**Summary:**

The submission provides a new view of Reinforcement Learning from Human Feedback (RLHF) objectives. This contribution is supported by theoretically establishing a unique normal form, learning guarantees and testing lower bounds, among others.

**Strengths:**

The submission provides a broad range of contributions; these are primarily foundational, but are also supported by a light-weight demo implementation.

**Weaknesses:**

The submission is not at all suited to the general ICLR community, as it lacks any sort of introduction to the studied area. Naturally, not every ICLR paper needs to provide contributions that are central to everyone in the conference's community, but as a general rule I would expect that at least the central concepts and contributions are explained on a level that would be understandable to at least a reasonable portion of the attendees. Yet in this particular case, it feels as if the authors have intentionally omitted anything that would help non-experts understand the submission's contributions. There is no general introduction to the topic - in fact, even the central abbreviation (RLHF) is never explained. All the discussion of prior work is concentrated in a half-page Related Work section on page 8, which just summarizes the main take-aways from previous articles but does not explain the general research direction or provide any context. Apart from that, the whole summary of the research area and desiderata is concentrated into the first two essentially empty sentences in the Introduction. Central concepts are not introduced with sufficient rigor and assume knowledge with concepts such as "margins" without providing any references for where a reader could at least theoretically obtain the necessary background.

I have no idea why the authors chose to present their results with this style, but it makes the submission essentially useless to all but the few experts who work on RLHF. The way the manuscript entirely ignores the need to convey information to researchers outside of its specific sub-area contrasts not only the general structure of ICLR submissions in other subfields, but also the much more welcoming style employed in many of the previous works on the topic cited by the submission. In its current form, I simply cannot see the submission as being ready to appear in the ICLR proceedings.

**Questions:**

None, but the authors are welcome to respond to the weaknesses listed above.

---

### Official Review · Reviewer_hps1 · 2025-10-28

**Soundness:** 1
**Presentation:** 1
**Contribution:** 1
**Rating:** 0
**Confidence:** 4

**Summary:**

Honestly, I could not understand what this paper is about -- see the weaknesses section

**Strengths:**

once again, it is hard to judge strengths of the paper in the current form, see weaknesses section.

**Weaknesses:**

The paper lacks many components that are expected from an ICLR submission -- motivation, introduction for non-specialists, proper discussion and comparison with the existing work. The introduction is rather short and right away starts operating with terminology like RLHF, ladders, margins and so on, which is never properly defined (for instance, it is not even defined what RLHF stands for). As a result, I am not even sure to which area this paper belongs to... If I'm not mistaken, there are no references until page 8, where a rather breve survey of the existing work appears (which looks more like a list of key words with references).

As a summary, the current submission looks at best like a draft, but it is definitely not ready for a publication.

**Questions:**

no questions at this stage

---

### Official Review · Reviewer_xc5G · 2025-11-05

**Soundness:** 2
**Presentation:** 1
**Contribution:** 1
**Rating:** 0
**Confidence:** 4

**Summary:**

This paper considers the question of determining when two objective functions used in RLHF are equivalent. It takes a formal verification approach, and the main result appears to be a canonicalizing algorithm for RLHF objectives.

Unfortunately the paper is written in a completely impenetrable way and is unsuitable for publication in this state.

**Strengths:**

N/A

**Weaknesses:**

The paper is written in an highly impenetrable and obscure fashion, seemingly with extensive usage of an LLM. The problem is never properly defined, motivation is barely provided, terminology is used without explanation, exposition and organization is absent, and the writing is extremely terse and superficially mathy to the point of deliberate obscurantism. Sample the first sentence of Section 3: "This section formalizes an equational theory for RLHF ladders and shows that, within the reducible class R (Assumptions (R1)–(R3) in Preliminaries), a terminating and locally confluent rewrite system yields a unique normal form (up to a fixed gauge)".

Regardless of contribution, the paper simply cannot be read in this state, much less accepted for publication.

**Questions:**

If the authors are serious about their contribution, I would recommend rewriting the paper entirely from scratch and making it accessible to a general ML audience.

---

### Official Review · Reviewer_caKT · 2025-11-05

**Soundness:** 3
**Presentation:** 3
**Contribution:** 3
**Rating:** 6
**Confidence:** 3

**Summary:**

The paper uses tools from logic and rewrite theory to show that many RLHF objectives are algebraically equivalent and can be canonicalized in a certain way. For irreducible methods, the theory can produce finite witnesses that show irreducibility.

**Strengths:**

The theory is novel and interesting. It is also correct as far as the reviewer can tell. The paper is comprehensive and cover most of the questions that could be asked in this line of work, with limitations clearly explained. Examples are performed on latest methods of RLHF, and key takeaways from each sections are cleanly stated.

**Weaknesses:**

While expected from a more algebraic approach, the main drawbacks of the paper include:
1. Lack of immediate actionable takeaway: At the heart of the paper is a reduction technique to rewrite one method of RLHF to another, base around preserving the margin. However, this does not necessarily mean that one method is preferred to another, or that the canonical method is superior to reducible methods that it is reducing to. With their reduction, the authors are able to canonicalize many ideas, from margin to regrets, etc, but it would be more valuable to demonstrate, at least experimentally why a canonicalized version would be preferred (e.g. numerical stability).
2. Margin-equivalent may obfuscate finer details in optimization and generalization: the canonicalization that serves as one of the key contributions of the paper is based around defining methods to be equivalent when they are (Bayes) margin-equivalent. However, it is known from margin theory of classification optimization that the value of the margin induced by the hypothesis function has impact on, say, implicit biases of gradient descents (e.g. Lyu and Li 2019 "Gradient Descent Maximizes the Margin of Homogeneous Neural Networks" or Ji and Telgarsky 2019, 2020 "The implicit bias of gradient descent on nonseparable data", etc.). There are also margin-based generalization bounds (Bartlett, Foster and Telgarsky 2017 "Spectrally-normalized margin bounds for neural networks", etc.). While these works are for supervised classification task, it highlights the fact that even if two methods produce the same margin, they may have different optimization and generalization properties altogether.
3. Guarantees are rather loose (which is expected from a more algebraic/general approach): for instance, the regret transfer bound is only guaranteed under some nondecreasing phi, which can be quite large.

**Questions:**

1. Does the proposed canonicalization works for binary classification in the supervised setting?
2. Can you use this method to canonicalize deep learning architectures, instead of just lose functions? Perhaps easier, can you apply this to different regularizers?

---

### Meta-Review · Area_Chair_Dzdy · 2026-01-04

**Summary:**

Reviewers raised several crucial concerns regarding the presentation, motivation, and significance of the paper. As there was no rebuttal from the authors, these concerns remain unaddressed. I therefore recommend rejection of the paper.

**Reviewer Concerns:**

Since there was no rebuttal from the authors, the reviewers’ concerns remain outstanding.

**Reviewer Scores:**

Since there was no rebuttal from the authors, active discussion between the authors and reviewers is unlikely to occur. Therefore, the reviewers are unlikely to change their scores.

---

### Decision · Program_Chairs · 2026-01-26

Reject